# RESfM: Robust Deep Equivariant Structure from Motion

**Fadi Khatib**[1][*]  **Yoni Kasten**[2]  **Dror Moran**[1]  **Meirav Galun**[1]  **Ronen Basri**[1]
[1]Weizmann Institute of Science  [2]NVIDIA

## ABSTRACT

Multiview Structure from Motion is a fundamental and challenging computer vision problem. A recent deep-based approach utilized matrix equivariant architectures for simultaneous recovery of camera pose and 3D scene structure from large image collections. That work, however, made the unrealistic assumption that the point tracks given as input are almost clean of outliers. Here, we propose an architecture suited to dealing with outliers by adding a multiview inlier/outlier classification module that respects the model equivariance and by utilizing a robust bundle adjustment step. Experiments demonstrate that our method can be applied successfully in realistic settings that include large image collections and point tracks extracted with common heuristics that include many outliers, achieving state-of-the-art accuracies in almost all runs, superior to existing deep-based methods and on-par with leading classical (non-deep) sequential and global methods.

## 1 INTRODUCTION

Simultaneous recovery of camera pose and 3D structure from large image collections, commonly termed Multiview Structure from Motion (SfM), is a longstanding fundamental problem in computer vision with applications in augmented and virtual reality, robot manipulation, and more. Classical, as well as some recent deep-based SfM techniques, rely on extracting point tracks, i.e., point matches across multiple frames, to solve for camera poses and structure. However, existing heuristics for point track extraction and chaining often return erroneous (*outlier*) point matches due to significant viewpoint, illumination differences, and repetitive scene structures, adversely affecting the performance of SfM methods. Designing robust methods that can effectively overcome the effect of such outliers, therefore, has been an active thread of research in the past several decades.

A promising approach to SfM is based on generalizations of *projective factorization*. This approach uses the observation that if we stack the tracked points in a matrix, which we denote by $M$, and assuming $M$ has no missing entries and is error-free, then there exists a set of scale factors such that scaling each tracked point in $M$ will make it rank 4. Enforcing this constraint allows us to recover both the 3D positions of the points that generate the tracks and the poses of the observing cameras (Sturm & Triggs, 1996). However, prior algorithms required $M$ to be complete and error-free; they were sensitive to initialization and were applied only in uncalibrated settings, producing a reconstruction up to a global projective ambiguity.

A deep network architecture for projective factorization was recently proposed in (Moran et al., 2021). Their network uses a sets-of-sets architecture (Hartford et al., 2018), which is permutation equivariant to both the rows and columns of the point track matrix. Brynte et al. (2023) proposed a related architecture that uses self-attention blocks to improve runtime further. Both these networks showed promising results on image collections acquired with a single camera. However, as we show in this paper (see also Figure 1), they fail to handle collections of internet photos, primarily because they were not designed to remove outlier matches.

In this paper, we aim to construct a robust network for projective factorization that can handle outliers in realistic settings. To this end, we return to (Moran et al., 2021)'s architecture and enrich it with an outlier removal module. Our module is integrated into the equivariant architecture, allowing it to identify outliers if their motion is inconsistent with other points in the same image or if their motion is

---

[*]The project page: `https://robust-equivariant-sfm.github.io/`

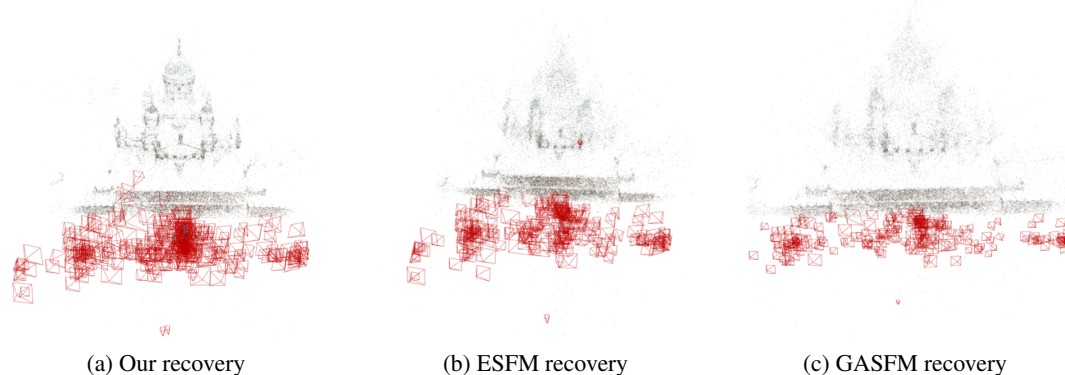

(a) Our recovery        (b) ESFM recovery        (c) GASFM recovery

Figure 1: **3D reconstruction and camera pose recovery in the presence of outliers.** The figure shows reconstruction results (point clouds) and camera poses (in red) obtained with our method (left) and existing deep-based methods including ESFM (Moran et al., 2021) (middle), and GASFM (Brynte et al., 2023) for scene 0185 (MegaDepth dataset, 30% outliers). It is evident that our method copes well with outliers in contrast to these existing methods.

at odds with the motion of other points in their track. We train this module with a cross-entropy loss, using labels that are inferred automatically using COLMAP reconstructions, allowing for cross-scene generalization. We further utilize a final bundle adjustment (BA) step that is robust to classification errors. Experiments show that our network improves over the methods of Moran et al. (2021); Brynte et al. (2023) in almost all runs, obtaining state-of-the-art accuracies and runtimes comparable to the leading classical methods, including COLMAP, Theia, and GLOMAP (Schönberger & Frahm, 2016; Sweeney et al., 2015; Pan et al., 2024).

In summary, our contributions are:

1. A deep network for *robust* multiview SfM, utilizing a sets-of-sets permutation equivariant architecture.
2. A trainable architecture allows for cross-scene and cross-dataset generalization.
3. Our method is applied to large collections (hundreds of images) of uncontrolled internet photos. This, to the best of our knowledge, is the first deep method for simultaneous recovery of structure and camera pose that handles such challenging inputs.
4. Our method achieves highly accurate recovery of camera pose and structure, superior to existing deep methods and on par both in accuracy and speed with state-of-the-art classical methods.
5. A benchmark with point tracks and pseudo ground truth computed with COLMAP on the MegaDepth and 1DSFM datasets.

## 2   RELATED WORK

The recovery of camera pose and 3D structure from image collections has been a central subject of research in computer vision in the past several decades, leading to multiple breakthroughs that enabled accurate reconstructions from hundreds and even thousands of unordered images (Agarwal et al., 2011; Schönberger & Frahm, 2016; Snavely et al., 2006; Wu, 2013). In the typical *sequential pipeline* point tracks are first extracted from the input images. Camera pose and 3D structure are next computed for two images and then updated by processing the remaining images one by one. This pipeline yields highly accurate recovery of pose and structure but can be slow when applied to large image collections. Alternative global approaches attempt to compute camera poses by a process of "averaging" the relative rotations and translations estimated from pairwise essential matrices (Martinec & Pajdla, 2007; Özyeşil et al., 2017; Kasten et al., 2019). The recent Theia (Sweeney et al., 2015) and GLOMAP (Pan et al., 2024), in particular, were shown to yield accurate recovery. Related to our approach is the projective factorization (PF) method (Sturm & Triggs, 1996; Dai et al., 2010; Lin et al., 2017), which is based on the observation that the full track matrix is derived from a rank four matrix by a per-point scale factor. Classical PF algorithms, however, are limited to uncalibrated settings and address neither missing entries nor outliers.

Recent deep-learning techniques attempt to improve these pipelines by exploiting priors over the input images, camera settings, and 3D structures learned with these networks. Existing methods attempt to improve keypoint detection and matching (Lindenberger et al., 2021; Sun et al., 2021; DeTone et al., 2018; Lindenberger et al., 2023; Sarlin et al., 2020), produce point tracks (He et al., 2023), formulate differentiable alternatives to RANSAC (Yi et al., 2018; Zhang et al., 2019; Zhao et al., 2021; Sun et al., 2020), or directly infer the relative orientation and location for pairs of images (Khatib et al., 2024; Laskar et al., 2017; Cai et al., 2021; Rockwell et al., 2022; Arnold et al., 2022).

Recent work proposed end-to-end methods for camera pose estimation. Works such as RelPose (Zhang et al., 2022) and its successor, RelPose++ (Lin et al., 2023), harness energy-based models to recover camera poses from inputs that include the relative rotations between images. Similarly, SparsePose (Sinha et al., 2023) learns to regress initial camera poses which are then refined iteratively. PoseDiffusion (Wang et al., 2023b) employs a diffusion model to refine camera poses. Zhang et al. (2024) improves on this by focusing on purifying the camera rays. These works, however, are only applied to small collections of images (typically $\leq 30$) and are commonly applied in object-centric scenarios, e.g., as in (Reizenstein et al., 2021). Recent learnable SfM pipelines such as VGGSfM (Wang et al., 2023a), DUST3R (Wang et al., 2023c), and MAST3R (Leroy et al., 2024) are still limited to handling only a small number of images, while Ace-Zero (Brachmann et al., 2024) and FlowMap (Smith et al., 2024) are applicable to video sequences with constant illumination.

Motivated by projective factorization schemes Moran et al. (2021) presented a trainable network for simultaneous camera pose and 3D structure recovery. Arranging the input point tracks in a matrix, an equivariant network to row and column permutations, i.e., sets-of-sets architecture, is used to regress the camera poses and the 3D point cloud coordinates. The network is trained with unsupervised data by optimizing a reprojection loss and is applied at inference to novel scenes not seen in training. Additional fine-tuning and Bundle Adjustment (BA) are applied to attain sub-pixel reprojection errors. This approach achieved highly accurate pose and structure recovery on large image collections (with several hundreds of images) of outdoor urban scenes. However, it has a significant limitation as it requires the point tracks matrix, to be almost free of outliers. This greatly restricts its applicability in realistic settings.

GASFM (Brynte et al., 2023) replaces the set-of-sets architecture in (Moran et al., 2021) with a graph attention network for increased expressiveness, enabling them to avoid fine-tuning, thereby reducing inference runtime compared to (Moran et al., 2021) without compromising the performance. Chen et al. (2024) applies (Moran et al., 2021)'s network (with some modifications) to aerial images captured with GPS information under roughly constant camera orientation. Neither Brynte et al. (2023) nor Chen et al. (2024) demonstrate results on point tracks contaminated by a significant portion of outliers.

Our approach extends and improves over the work of Moran et al. (2021) by integrating a multiview inlier/outlier classification module that respects and exploits the equivariant structure of the network and by introducing a robust BA scheme. This allows us to work with realistic point tracks contaminated with outliers obtained with standard heuristics and still achieve high-accuracy performance.

## 3 METHOD

### 3.1 PROBLEM FORMULATION

We assume a stationary scene viewed by $m$ cameras with unknown poses. We obtain as input a (sparse) point track tensor $M$ that includes 2D observations of $n$ 3D points viewed by partial sets of cameras. We further assume in this work that the cameras are internally calibrated. A camera matrix therefore is expressed in the form of $P_i = [R_i | \mathbf{t}_i]$, where $R_i \in SO(3)$ is a rotation matrix. With this notation, the camera is placed at the position $-R_i^T \mathbf{t}_i$. We denote by $\mathbf{X}_j \in \mathbb{R}^3$ the $j$'th 3D scene point and by $\mathbf{x}_{ij} \in \mathbb{R}^2$ its observed position in the $i^{\text{th}}$ image. The set $T_j = \{\mathbf{x}_{i_1 j}, \mathbf{x}_{i_2 j}, ...\}$ with $C_j = \{i_1, i_2, ...\} \subseteq [m]$ represents the $j$'th track, associated with the $j$'th 3D point. These tracks are generally constructed in a preprocessing step by employing heuristics and, therefore, are contaminated by small displacement errors (noisy measurements) and outliers.

We arrange the tracks $T_1, \ldots, T_n$ in the columns of the measurement point track tensor $M$. The tensor $M$ is of size $m \times n \times 2$, so the rows of $M$ correspond to the $m$ cameras, and its columns correspond

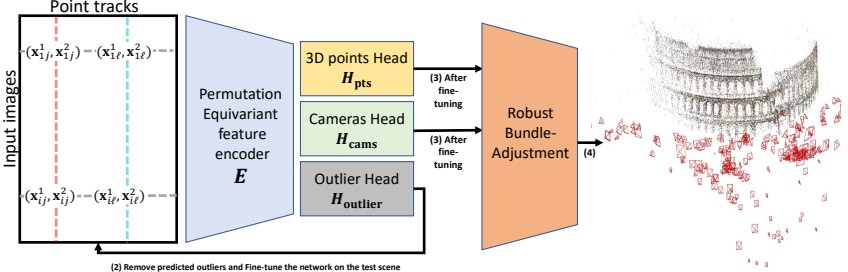

Figure 2: **Network architecture**. Our network comprises four main steps: (1) Given a point track tensor (left), an equivariant feature encoder module (light blue) produces a latent feature representation. (2) The outlier head (gray) applies an inlier/outlier classification. Predicted outliers are then removed from the input track, and the encoder is fine-tuned for 1000 epochs. (3) 3D point locations and camera poses are predicted by the point head (yellow) and the camera head (green), respectively. (4) The predicted camera poses and the point cloud are finally optimized by robust bundle adjustment (orange).

to the unknown 3D points whose contaminated projections are given. We set $(M_{ij1}, M_{ij2}) \doteq \mathbf{x}_{ij}$ if $i \in C_j$ and otherwise leave $M_{ij1}$ and $M_{ij2}$ empty.

We aim to recover the pose parameters of the $m$ camera matrices and the 3D positions of the $n$ feature points. Additionally, we seek to classify each track point as either an inlier or an outlier. The output of our network will comprise three matrices: $P \in \mathbb{R}^{m \times 7}$ for the set of cameras matrices, $X \in \mathbb{R}^{n \times 3}$ for the 3D structure, and $O \in \mathbb{R}^{m \times n}$ for the track point classification results. We represent camera pose by a rotation quaternion $\mathbf{q}_i \in \mathbb{R}^4$ and a translation vector $\mathbf{t}_i \in \mathbb{R}^3$. Classification results in $O$ are represented by scores in $[0, 1]$, where $\mathbf{x}_{ij}$ with $O_{ij} \geq 0.6$ is considered an outlier. (The threshold 0.6 was determined by a hyperparameter search.) Note that we classify individual points as outliers, while other points in the same track may be classified as inliers.

## 3.2 NETWORK ARCHITECTURE

We use an architecture that is equivariant to row and column permutations of the tensor $M$, equivalent respectively to independent permutations of the cameras and the 3D points. It comprises four modules: a permutation equivariant feature encoder, followed by an inlier/outlier classification module, pose, and structure regression modules (see Fig. 2).

**Permutation equivariant feature encoder.** This module takes as input the tensor $M \in \mathbb{R}^{m \times n \times 2}$ and outputs a latent representation in $\mathbb{R}^{m \times n \times d}$, where $d = 256$. It comprises three permutation equivariant layers with linear maps interspersed with pointwise non-linear ReLU function. A linear equivariant map is the core of permutation equivariant networks. In our case, permutation equivariance should apply to the rows (cameras) and columns (3D points) of the tensor $M$. Hartford et al. (2018) showed that the space of linear maps that keep the permutation equivariant property of a single channel (i.e., the input and the output are matrices) is spanned by the identity, the row sums, the column sums, and the matrix sum. This can be applied to multiple input and output channels from a feature space with $d$ channels $\mathbb{R}^{m \times n \times d}$ to a feature space with $d'$ channels $\mathbb{R}^{m \times n \times d'}$ in the following way:

$$L(\tilde{M})_{ij} = W_1 \tilde{M}_{ij} + W_2 \sum_{k=1}^{m} \tilde{M}_{kj} + W_3 \sum_{l=1}^{n} \tilde{M}_{il} + W_4 \sum_{k=1}^{m} \sum_{l=1}^{n} \tilde{M}_{kl} + \mathbf{b}, \tag{1}$$

where $\tilde{M}_{ij} \in \mathbb{R}^d$ represents the vector of entries of $\tilde{M}$ at the $(i, j)$ position, $W_i \in \mathbb{R}^{d' \times d}$ for $i = 1, ..., 4$, and $\mathbf{b} \in \mathbb{R}^{d'}$ are learnable parameters. We follow Moran et al. (2021) and replace sums by averages over the non-empty entries of $M$, yielding invariance to the number of observed projections.

**Inlier/Outlier classification module.** Given the latent representation $\mathbb{R}^{m \times n \times d}$ provided by the equivariant feature encoder, this module, $H_{\text{outlier}}$, employs three MLP layers followed by a sigmoid function. The output is a matrix $O \in [0, 1]^{m \times n}$ expressing the probability of each entry being an outlier. Note that this module respects the equivariance property, allowing our model to classify track

points based on their relations to other points of the same track as well as other points within the same image.

**Pose and structure regression modules.** Utilizing the latent representation $\mathbb{R}^{m \times n \times d}$ from the equivariant feature encoder, the pose and structure heads, $H_{\text{cams}}$ and $H_{\text{pts}}$, consist of three MLP layers each. The first head, $H_{\text{cams}}$, processes the pooled average of the features along the columns $\mathbb{R}^{m \times d}$, where each camera's features are encoded into a vector in $\mathbb{R}^d$, and outputs a tensor in $\mathbb{R}^{m \times 7}$. This tensor encapsulates the camera's translation vector (first three values) and orientation (last four values, represented as a quaternion), collectively forming the camera's projection matrix. The second head, $H_{\text{pts}}$, processes the pooled average of features along the rows $\mathbb{R}^{n \times d}$, where each scene point is encoded by a vector in $\mathbb{R}^d$, and outputs a tensor in $\mathbb{R}^{n \times 3}$, representing the coordinates of the scene points.

### 3.3 LOSS FUNCTION

Our loss function combines two terms

$$\mathcal{L} = \mathcal{L}_{\text{outliers}} + \alpha \mathcal{L}_{\text{reprojection}} \tag{2}$$

where the hyperparameter $\alpha = 10$ balances the two terms and was determined by a hyperparameter search. The outlier classification loss, $\mathcal{L}_{\text{outliers}}$, employs a Binary Cross-Entropy (BCE) loss. The second part of the loss $\mathcal{L}_{\text{reprojection}}$ is an unsupervised objective, and it includes reprojection and hinge loss terms. The reprojection loss aims to minimize the error between projected scene points and their detected positions in the image, similar to the objective in bundle adjustment. A hinge loss is also used to prevent the prediction of non-positive depth values, thus ensuring a physically plausible scene reconstruction.

$$\mathcal{L}_{\text{reprojection}} = \frac{1}{p} \sum_{i=1}^{m} \sum_{j=1}^{n} \xi_{ij} s_{ij}, \tag{3}$$

where $p = \sum_{j=1}^{n} |T_j|$ the number of measured projections, and $\xi_{ij} \in \{0, 1\}$ is indicating whether point $\mathbf{X}_j$ is detected in image $I_i$.

$$s_{ij} = \begin{cases} r_{ij}, & P_i^3 \mathbf{X}_j \geq h \\ h_{ij}, & P_i^3 \mathbf{X}_j < h, \end{cases}$$

and

$$r_{ij} = \left\| \left( \mathbf{x}_{ij}^1 - \frac{P_i^1 \mathbf{X}_j}{P_i^3 \mathbf{X}_j}, \mathbf{x}_{ij}^2 - \frac{P_i^2 \mathbf{X}_j}{P_i^3 \mathbf{X}_j} \right) \right\|,$$

and $h_{ij} = \max(0, h - P_i^3 \mathbf{X}_j)$, where $h > 0$ is a small constant (in our setting, $h = 0.0001$). $h_{ij}$ therefore is the hinge loss for the depth value of $\mathbf{X}_j$ in image $I_i$.

### 3.4 TRAINING

During training, we processed all training scenes sequentially for each epoch. For each scene, we randomly selected a subset of 10%-20% of the images. We employed a validation set for early stopping, selecting the checkpoint with minimal error. More details can be found in Appendix A.

### 3.5 INFERENCE

After training our network, we test the network on *unseen scenes*. We first apply inlier/outlier classification predictions to remove outliers, and then fine-tune the network for the tested scene. This yields predictions for camera poses and 3D point locations. Having those predictions, we apply our robust Bundle Adjustment procedure.

**Fine-tuning.** At this step, we leverage our inlier/outlier classification prediction and remove the points predicted as outliers from the point tracks tensor $M$ (using the threshold 0.6). Subsequently, similar to (Moran et al., 2021), we fine-tune the network for the tested scene by minimizing the unsupervised reprojection loss equation 3 for 1000 epochs.

**Robust Bundle Adjustment.** As common in SfM pipelines, a post-processing step of applying bundle adjustment is necessary to achieve sub-pixel reconstruction accuracy. Standard BA was found to be ineffective as it was unable to handle the remaining outliers. We therefore utilize a Robust Bundle Adjustment process:

1. Apply bundle adjustment (BA) to the predicted cameras and predicted 3D points.

2. Remove 3D points with projection error higher than 5 pixels and remove 3D points viewed in fewer than 3 cameras.

3. If the view graph of the point tracks becomes unconnected due to the removals, then we take the largest component and discard point tracks that originate from the removed images.

4. Triangulate the remaining point tracks and apply a second round of (standard) BA.

More details on the robust BA stage can be found in Appendix A.

## 4    EXPERIMENTS

### 4.1    DATASETS

Our network is trained on scenes from the MegaDepth dataset (Li & Snavely, 2018). It is then tested on both novel scenes from the MegaDepth dataset and in cross-dataset generalization tests on the 1DSfM dataset (Wilson & Snavely, 2014), Strecha (Strecha et al., 2008), and BlendedMVS (Yao et al., 2020). These datasets offer a diverse range of real-world scenes, which are instrumental in assessing the robustness and versatility of our proposed architecture across different environments and challenges.

The point tracks, which our network takes as input, are constructed by concatenating pairwise matches between images within each scene. These matches are obtained by applying RANSAC to SIFT matches, ensuring a robust selection of correspondences by mitigating the influence of outliers. The detailed procedure for constructing these point tracks, including parameter settings and algorithmic choices, is provided in Appendix C.

**MegaDepth (Li & Snavely, 2018).** The MegaDepth dataset comprises 196 different outdoor scenes, each populated with internet photos showcasing popular landmarks around the globe. To facilitate a comprehensive evaluation, we divide the dataset into two groups based on the number of images per scene: (1) scenes with fewer than 1000 images, and (2) scenes with more than 1000 images. From the first group, we randomly sampled 27 scenes to serve as our training dataset, along with four scenes designated for validation purposes. For the test set, we randomly selected 14 scenes from the first group. Moreover, from each scene in the second group, we randomly sampled 300 images to represent a condensed version of the scene. These samples form a part of the test dataset, introducing a significant challenge for Structure from Motion (SfM) methods due to the reduced number of images representing vast and complex scenes. In Table 1, the scenes above the middle rule belong to Group 1 of scenes with fewer than 1000 images, while the scenes below the rule belong to Group 2.

**1DSFM (Wilson & Snavely, 2014).** The 1DSFM dataset is renowned for its collection of diverse scenes reconstructed from community photo collections. It includes a variety of urban locations, making it a valuable resource for evaluating Structure from Motion (SfM) and multi-view stereo algorithms. For our purposes, the dataset offers a challenging yet realistic setting to evaluate our architecture's effectiveness in dealing with large-scale reconstructions. Specifically, we train the models on MegaDepth dataset and assess its generalization to 1DSFM dataset.

**Strecha (Strecha et al., 2008).** The Strecha dataset is widely used for benchmarking 3D reconstruction algorithms and consists of five outdoor scenes. It provides high-resolution images alongside ground-truth data acquired with a LIDAR system. However, each scene includes only a small ($\leq 30$) number of images acquired with a single camera. We perform our tests on four of these scenes.

**BlendedMVS (Yao et al., 2020).** The BlendedMVS is a synthetic dataset built by reconstructing textured meshes and rendering them into color images and depth maps, which are blended with the original inputs to create realistic data, thereby generating ground truth for camera poses.

**Ground truth camera poses.** Challenging datasets such as MegaDepth and 1DSFM lack ground truth. Therefore, as is common in the field Jiang et al. (2013); Wilson & Snavely (2014); Cui & Tan (2015); Ozyesil & Singer (2015); Brynte et al. (2023); Wang et al. (2023a); Zhang et al. (2024), we utilize COLMAP, a state-of-the-art incremental Structure from Motion (SfM) method, to establish ground truth camera poses for the scenes in those datasets. COLMAP stands as one of the most popular solutions in the field due to its robust performance in reconstructing 3D models from unordered image collections. We apply COLMAP directly to the scene images to obtain the ground truth poses. Additionally, we show results with the smaller datasets Strecha (Strecha et al., 2008) and BlendedMVS (Yao et al., 2020) for which ground truth camera poses are available.

## 4.2 BASELINES

**ESFM (Moran et al., 2021).** We compare our method to this equivariant SfM method where, for fairness, we replace BA with our robust BA. The model is trained on MegaDepth dataset and evaluated both on MegaDepth and 1DSFM. We conduct two distinct evaluations: one in which ESFM is trained and tested on the original point tracks, which include outliers (denoted as ESFM), and another evaluation where the model is trained and tested on the same point tracks, but in which outliers have been removed (this version is denoted as ESFM*). By testing ESFM on the original contaminated tracks, we evaluate the method's ability to handle realistic settings. A comparison with outlier-free tracks provides an empirical upper bound for our method.

**GASFM (Brynte et al., 2023).** We compare our method to GASFM, which replaces the set-of-sets architecture in ESFM (Moran et al., 2021) with a graph attention network. This modification increases expressiveness, enabling them to avoid fine-tuning and thereby reduce inference runtime compared to the traditional SfM method (Schönberger & Frahm, 2016), without compromising performance when tested on the outlier-free Olsson's dataset (Olsson & Enqvist, 2011). We trained and tested GASFM on the original MegaDepth point tracks (which include outliers) and also tested it on the 1DSFM dataset scenes.

To allow a fair comparison, in both ESFM and GASFM, the inference step is followed by 1000 epochs of fine-tuning.

**VGGSfM (Wang et al., 2024)** is a differentiable, end-to-end trainable SfM pipeline, simplifying some of its components to streamline the process while maintaining accurate 3D reconstruction capabilities.

**MASt3R (Leroy et al., 2024)** is a pipeline for SfM that merges pairwise pointmap predictions through an optimization-based global alignment procedure to handle image collections effectively.

**Theia (Sweeney et al., 2015).** A widely recognized global SfM pipeline that begins by estimating camera rotations using rotation averaging, followed by translation averaging to estimate camera positions. It concludes with global triangulation to reconstruct the 3D point cloud and a final bundle adjustment, similar to other global SfM techniques.

**GLOMAP (Pan et al., 2024).** GLOMAP is a newly proposed global SfM pipeline that addresses the limitations of previous global methods, which were considered efficient but less robust than incremental approaches. Instead of relying on separate translation averaging and point triangulation, GLOMAP combines them into a single global positioning step, optimizing both camera positions and 3D structure simultaneously.

## 4.3 METRICS AND EVALUATION

We evaluate our results using camera position and orientation errors. Specifically, after performing a similarity alignment per scene, we compare our camera orientation predictions with the ground truth ones by measuring angular differences in degrees. Similarly, we measure differences between our predicted and ground truth camera locations. For a fair comparison, both our method and all the baseline methods (except VGGSfM, which is applied directly to the input images) were run with the same set of point tracks. For all methods, we apply a final post-processing step of our robust bundle adjustment. When highlighting the best results in the tables, we do not compare to ESFM*.

Table 1: **MegaDepth experiment.** For each scene, we show the number of input images (denoted $N_c$) and the fraction of outliers. For each model, we show the number of images used for reconstruction (denoted $N_r$) and mean values of the rotation (in degrees) and translation errors. (Above the middle rule are Group 1 scenes with <1000 images; below are Group 2 scenes with >1000 images, subsampled to 300 for testing.) Winning results are marked in bold and underlined. Yellow represents the best result among the deep-based algorithms and green among the classical algorithms.

| Scene | $N_c$ | Outliers% | Ours | | | ESFM | | | GASFM | | | Theia | | | GLOMAP | | | ESFM* | | |
|---|---|---|---|---|---|---|---|---|---|---|---|---|---|---|---|---|---|---|---|---|
| | | | $N_r$ | Rot | Trans | $N_r$ | Rot | Trans | $N_r$ | Rot | Trans | $N_r$ | Rot | Trans | $N_r$ | Rot | Trans | $N_r$ | Rot | Trans |
| 0238 | 522 | 44.6 | 283 | 2.61 | **0.325** | 76 | 11.49 | 1.088 | 76 | 9.89 | 0.969 | **506** | 1.21 | 0.334 | 499 | **0.74** | 0.349 | 512 | 3.06 | 0.142 |
| 0060 | 528 | 41.6 | 503 | 0.29 | **0.029** | 303 | 14.92 | 2.167 | 258 | 18.48 | 2.736 | **525** | 0.85 | 0.124 | 522 | **0.11** | 0.048 | 524 | 0.02 | 0.005 |
| 0197 | 870 | 40.7 | 667 | 4.22 | 0.333 | 281 | 9.62 | 0.980 | 454 | 12.23 | 1.678 | **855** | 1.16 | 0.227 | 814 | **0.43** | **0.129** | 825 | 0.41 | 0.050 |
| 0094 | 763 | 40.1 | 537 | 3.77 | 0.750 | 93 | 14.91 | 1.772 | 359 | **2.27** | **0.322** | **742** | 0.75 | 0.160 | 717 | 0.88 | 3.907 | 643 | 4.55 | 1.018 |
| 0265 | 571 | 38.8 | 346 | **1.25** | **0.389** | 270 | 22.14 | 2.712 | 274 | 22.29 | 2.756 | 554 | 5.83 | 2.216 | **558** | 7.46 | 2.839 | 559 | 0.12 | 0.039 |
| 0083 | 635 | 31.3 | 596 | 0.64 | 0.058 | 568 | 4.40 | 0.558 | 574 | 3.95 | 0.245 | **632** | 0.37 | 0.372 | 614 | **0.08** | **0.016** | 556 | 21.02 | 1.904 |
| 0076 | 558 | 30.5 | 524 | 0.37 | 0.094 | 454 | 3.86 | 0.655 | 431 | **0.11** | **0.015** | **549** | 0.78 | 0.120 | 541 | 0.17 | 0.042 | 547 | 0.04 | 0.006 |
| 0185 | 368 | 30.0 | 350 | **0.06** | **0.010** | 261 | 1.76 | 0.271 | 254 | 1.36 | 0.184 | **365** | 0.41 | 0.094 | **365** | 0.16 | 0.051 | 130 | 38.49 | 2.478 |
| 0048 | 512 | 24.2 | 474 | 4.69 | 0.178 | 469 | 4.03 | 0.235 | 481 | 1.44 | **0.073** | **507** | 0.41 | 0.105 | 506 | **0.15** | 0.224 | 508 | 0.09 | 0.006 |
| 0024 | 356 | 23.0 | 309 | 2.03 | 0.398 | 271 | 9.08 | 1.320 | 300 | 9.80 | 2.546 | **355** | 0.56 | 0.219 | 339 | **0.15** | **0.104** | 343 | 2.78 | 0.568 |
| 0223 | 214 | 17.0 | 204 | 3.76 | 0.510 | 191 | 11.97 | 2.272 | 194 | 13.69 | 2.649 | 212 | 3.34 | 0.519 | 214 | **1.75** | **0.275** | 211 | 0.09 | 0.017 |
| 5016 | 28 | 0.2 | 28 | 0.12 | 0.016 | **28** | 0.09 | **0.015** | **28** | 0.09 | **0.015** | **28** | 0.10 | 0.061 | **28** | **0.08** | 0.046 | 28 | 0.04 | 0.009 |
| 0046 | 440 | 14.6 | 399 | 0.95 | 0.043 | 97 | 1.68 | 0.082 | 426 | 0.45 | 0.019 | 434 | 0.25 | 0.112 | **440** | **0.03** | **0.007** | 33 | 37.09 | 1.387 |
| 0099 | 299 | 47.4 | 190 | 3.53 | 0.709 | 104 | 4.17 | 0.862 | 128 | 8.13 | 1.116 | **297** | 3.28 | 0.664 | 255 | **0.15** | **0.085** | 243 | 6.22 | 1.075 |
| 1001 | 285 | 43.9 | 251 | **1.70** | **0.661** | 241 | 4.40 | 1.846 | **261** | 3.26 | 1.143 | 276 | 7.97 | 4.014 | **281** | 4.56 | 3.817 | 280 | 0.12 | 0.085 |
| 0231 | 296 | 42.2 | 246 | 0.84 | **0.065** | 214 | 0.85 | 0.080 | 209 | 0.91 | 0.088 | **286** | 1.37 | 0.322 | 279 | **0.73** | 0.134 | 284 | 0.56 | 0.022 |
| 0411 | 299 | 29.9 | 273 | **0.13** | **0.020** | 188 | 15.89 | 1.650 | 232 | 2.95 | 0.304 | **293** | 0.39 | 0.196 | 269 | 0.19 | 0.148 | 288 | 0.08 | 0.013 |
| 0377 | 295 | 27.5 | 210 | 0.29 | 0.018 | 162 | 0.54 | 0.044 | 167 | **0.13** | **0.013** | **269** | 1.13 | 0.205 | 268 | 0.65 | 0.237 | 279 | 1.19 | 0.147 |
| 0102 | 299 | 25.8 | 284 | 0.28 | **0.059** | 255 | 1.55 | 0.403 | 278 | 1.79 | 0.478 | **294** | 2.31 | 0.698 | 293 | **0.15** | 0.101 | 155 | 21.00 | 3.470 |
| 0147 | 298 | 24.6 | 207 | **4.62** | **0.325** | 197 | 4.90 | 0.522 | 225 | 10.89 | 0.961 | 284 | 6.36 | 0.934 | **290** | 6.75 | 3.542 | 251 | 3.22 | 0.215 |
| 0148 | 287 | 24.6 | 197 | **0.60** | **0.035** | 206 | 1.64 | 0.133 | 209 | 1.73 | 0.135 | 275 | 13.98 | 1.558 | **283** | 22.73 | 2.646 | 249 | 0.94 | 0.083 |
| 0446 | 298 | 22.1 | 288 | 0.72 | **0.046** | 283 | 2.02 | 0.193 | **291** | 1.71 | 0.237 | 289 | 1.23 | 0.391 | **296** | 0.20 | 0.071 | 294 | 0.92 | 0.115 |
| 0022 | 297 | 21.2 | 274 | 0.29 | **0.039** | 241 | 1.38 | 0.184 | 263 | 0.54 | 0.082 | **296** | 0.58 | 0.160 | 281 | **0.22** | 0.087 | 289 | 0.30 | 0.646 |
| 0327 | 298 | 21.0 | 271 | 0.26 | 0.090 | 281 | 1.83 | 0.398 | 284 | **0.25** | **0.029** | 288 | 1.27 | 0.360 | **290** | 15.54 | 2.035 | 294 | 0.06 | 0.008 |
| 0015 | 284 | 20.6 | 215 | 1.04 | 0.167 | 142 | 5.00 | 0.920 | 149 | 14.02 | 2.105 | 244 | 2.21 | 0.389 | **274** | **0.28** | **0.095** | 185 | 4.51 | 0.941 |
| 0455 | 298 | 19.8 | 293 | 0.68 | 0.105 | 293 | **0.68** | 0.138 | **294** | 0.74 | 0.144 | 294 | 0.77 | 0.159 | **298** | 0.35 | **0.064** | 298 | 0.89 | 0.109 |
| 0496 | 297 | 19.2 | 281 | **0.35** | **0.055** | 281 | 3.59 | 0.311 | 277 | 0.68 | 0.061 | 285 | 1.40 | 0.550 | **291** | 0.44 | 0.303 | 293 | 0.29 | 0.028 |
| 1589 | 299 | 17.4 | 290 | 0.14 | **0.019** | 284 | 0.92 | 0.131 | 283 | 2.71 | 0.505 | 288 | 0.82 | 0.193 | **299** | **0.07** | 0.041 | 296 | 0.40 | 0.053 |
| 0012 | 299 | 16.3 | 287 | **0.40** | **0.027** | 291 | 5.20 | 0.327 | **294** | 0.88 | 0.114 | 129 | 1.04 | 0.318 | **295** | 0.51 | 0.121 | 294 | 0.47 | 0.044 |
| 0104 | 284 | 16.2 | 193 | **0.29** | **0.029** | 220 | 24.40 | 2.174 | 228 | 4.10 | 0.306 | 265 | 17.05 | 1.530 | **280** | 19.69 | 0.834 | 200 | 0.78 | 0.044 |
| 0019 | 299 | 15.4 | 250 | **0.06** | **0.008** | 267 | 9.34 | 0.329 | 293 | 2.34 | 0.113 | 271 | 0.81 | 0.250 | **296** | 0.09 | 0.025 | 296 | 4.90 | 0.180 |
| 0063 | 293 | 14.5 | 262 | 0.46 | 0.048 | **262** | 2.15 | 0.456 | 257 | **0.44** | **0.040** | 268 | 0.92 | 0.605 | **288** | 0.32 | 0.100 | 275 | 0.32 | 0.301 |
| 0130 | 285 | 14.4 | 192 | **0.20** | **0.023** | 192 | 1.46 | 0.058 | **194** | 2.27 | 0.070 | 187 | 1.20 | 0.349 | **281** | 2.00 | 0.909 | 282 | 1.59 | 0.179 |
| 0080 | 284 | 12.9 | 139 | **0.59** | **0.096** | 137 | 1.34 | 0.325 | 139 | 2.18 | 0.104 | 278 | 2.62 | 0.868 | **283** | 1.92 | 0.237 | 163 | 1.92 | 0.173 |
| 0240 | 298 | 11.9 | 275 | 3.13 | 0.265 | 274 | **1.69** | **0.170** | 272 | 2.54 | 0.227 | 278 | 1.31 | 0.470 | **294** | **0.39** | **0.135** | 296 | 0.27 | 0.111 |
| 0007 | 290 | 11.7 | 172 | 0.91 | 0.041 | 260 | 38.74 | 2.284 | **280** | 1.87 | 0.101 | 277 | 1.24 | 0.174 | **290** | **0.19** | **0.035** | 286 | 1.59 | 0.264 |
| Mean | 379 | 25.6 | 298 | **1.29** | **0.169** | 239 | 6.77 | 0.780 | 267 | 4.53 | 0.630 | 346 | 2.42 | 0.556 | 353 | 2.51 | 0.662 | 319 | 4.45 | 0.443 |

## 4.4 RESULTS

Our results for the MegaDepth and 1DSFM test scenes are shown in Tables 1 and 2, respectively. Each table lists the number of input images ($N_c$), the fraction of outlier track points, and our results compared to the baseline methods. For each method, we provide the number of registered cameras ($N_r$) and mean camera orientation and position errors. It can be seen that our method outperforms both ESFM and GASFM in almost all runs, achieving accurate results that are close to what is obtained with ESFM on clean tracks (ESFM*). Our results are also on par with state-of-the-art classical methods, including Theia and GLOMAP, often yielding superior translation recovery (but handling fewer images). This demonstrates the utility of our method in realistic settings.

We further tested our method on the smaller Strecha and BlendedMVS datasets which include ground truth measurements. Our method is more accurate than the deep-based VGGSfM and MASt3R (that cannot run on the larger datasets) and is on par with the classical methods, see Tables 3 and 4.

**Qualitative results.** Figure 1 shows an example of 3D reconstructions and camera parameters obtained using our method compared to those obtained with ESFM and GASFM. These results clearly demonstrate that our method produces superior 3D reconstructions, effectively handling outliers in contrast to the other baselines. Additional qualitative results are provided in Appendix A.

**Runtime and resources.** Table 5 compares the runtimes of our method with the classical COLMAP, Theia, and GLOMAP when applied to the point tracks generated in our preprocessing stage. Our approach is significantly faster than COLMAP and GLOMAP but is slower than Theia. We note that

Table 2: **1DSFM experiment.** For each scene, we show the number of input images (denoted $N_c$) and the fraction of outliers. For each model, we show the number of images used for reconstruction (denoted $N_r$) and mean values of the rotation (in degrees) and translation errors. Winning results are marked in bold and underlined. Yellow represents the best result among the deep-based algorithms and green among the classical algorithms.

| Scene | $N_c$ | Outliers% | Ours $N_r$ | Rot | Trans | ESFM $N_r$ | Rot | Trans | GASFM $N_r$ | Rot | Trans | Theia $N_r$ | Rot | Trans | GLOMAP $N_r$ | Rot | Trans | ESFM* $N_r$ | Rot | Trans |
|---|---|---|---|---|---|---|---|---|---|---|---|---|---|---|---|---|---|---|---|---|
| Alamo | 573 | 32.6 | 484 | 3.66 | 0.515 | 457 | 4.53 | **0.319** | 448 | 5.85 | 0.460 | 553 | 4.42 | 1.433 | 557 | **2.45** | 1.520 | 526 | 3.17 | 0.321 |
| Ellis Island | 227 | 25.1 | 214 | 0.82 | **0.122** | 196 | 21.13 | 2.053 | 198 | 21.91 | 2.081 | 213 | 5.01 | 1.527 | 219 | **0.58** | 0.155 | 220 | 0.38 | 0.033 |
| Madrid Metropolis | 333 | 39.4 | 244 | 8.42 | 0.827 | 151 | 19.56 | 1.946 | 159 | 21.97 | 2.205 | - | - | - | 320 | **1.22** | **0.242** | 290 | 25.25 | 2.664 |
| Montreal Notre Dame | 448 | 31.7 | 346 | 2.82 | 0.352 | 309 | 10.13 | 1.773 | 311 | 11.77 | 1.557 | 422 | 4.47 | 1.285 | 444 | **0.60** | **0.211** | 414 | 0.16 | 0.020 |
| Notre Dame | 549 | 35.6 | 517 | **1.2** | **0.231** | 487 | 1.95 | 0.226 | 499 | 1.74 | 0.232 | 314 | 3.70 | 0.828 | 543 | 2.73 | 0.389 | 528 | 0.72 | 0.051 |
| NYC Library | 330 | 33.6 | 224 | 3.96 | 0.429 | 177 | 4.42 | 0.468 | 218 | 7.65 | 0.667 | 534 | 4.06 | 1.141 | 323 | **0.58** | **0.189** | 301 | 2.62 | 0.226 |
| Piazza del Popolo | 336 | 33.1 | 249 | 2.20 | **0.186** | 204 | 10.77 | 0.997 | 198 | 9.05 | 1.063 | 325 | 3.31 | 1.053 | 331 | **0.80** | 0.188 | 303 | 4.31 | 0.604 |
| Tower of London | 467 | 27.0 | 94 | **0.67** | **0.026** | 196 | 22.02 | 2.239 | 152 | 32.36 | 2.306 | 448 | 6.61 | 1.189 | 466 | 0.81 | 0.138 | 213 | 13.08 | 0.704 |
| Vienna Cathedral | 824 | 31.4 | 479 | **1.52** | **0.112** | 551 | 6.52 | 0.537 | 558 | 7.98 | 0.573 | 772 | 12.25 | 1.663 | 822 | 2.00 | 2.414 | 536 | 1.51 | 0.069 |
| Yorkminster | 432 | 29.0 | 331 | 14.54 | 1.468 | 215 | **10.63** | **0.637** | 251 | 13.01 | 0.846 | 390 | 8.35 | 1.916 | 418 | **0.95** | **0.316** | 389 | 12.63 | 0.994 |

Table 3: **Strecha experiment.** For each scene, we present the number of input images (denoted $N_c$) and the fraction of outliers. For each model, we show the number of images used for reconstruction (denoted $N_r$) and the mean values of the rotation error (in degrees), translation error (in meters) and runtime (in seconds). The best results are marked in bold and the second best are underlined.

| Scene | $N_c$ | Out.% | Ours $N_r$ | Rot | Trans | Time | MASt3R $N_r$ | Rot | Trans | Time | VGGSfM $N_r$ | Rot | Trans | Time | Theia $N_r$ | Rot | Trans | Time | COLMAP $N_r$ | Rot | Trans | Time | GLOMAP $N_r$ | Rot | Trans | Time |
|---|---|---|---|---|---|---|---|---|---|---|---|---|---|---|---|---|---|---|---|---|---|---|---|---|---|---|
| entry-P10 | 10 | 4.8 | 10 | 0.024 | 0.008 | 3.3 | 10 | 0.442 | 0.055 | 19 | 10 | 0.165 | 0.056 | 10.3 | 10 | 0.024 | 0.008 | 0.9 | 10 | 0.023 | 0.007 | 36.0 | 10 | 0.187 | 0.026 | 12.5 |
| fountain-P11 | 11 | 1.4 | 11 | 0.028 | 0.003 | 5.3 | 11 | 0.160 | 0.026 | 22 | 11 | 0.172 | 0.016 | 15.4 | 11 | 0.027 | 0.002 | 1.5 | 11 | 0.027 | 0.003 | 37.0 | 11 | 0.194 | 0.022 | 38.6 |
| Herz-Jesus-P8 | 8 | 1.8 | 8 | 0.026 | 0.004 | 2.6 | 8 | 0.363 | 0.037 | 16 | 8 | 0.206 | 0.042 | 8.7 | 8 | 0.025 | 0.005 | 0.6 | 8 | 0.026 | 0.004 | 22.0 | 8 | 0.091 | 0.015 | 5.0 |
| Herz-Jesus-P25 | 25 | 2.8 | 24 | 0.030 | 0.006 | 9.4 | 25 | 0.869 | 0.057 | 81 | 25 | 0.158 | 0.046 | 19.6 | 25 | 0.026 | 0.006 | 2.4 | 25 | 0.028 | 0.006 | 60.0 | 25 | 0.138 | 0.013 | 76.6 |
| Mean | 1.5 | 2.7 | 13.5 | 0.027 | 0.005 | 5.2 | 13.5 | 0.459 | 0.044 | 34.5 | 13.5 | 0.175 | 0.040 | 13.5 | 13.5 | 0.026 | 0.005 | 1.4 | 13.5 | 0.026 | 0.005 | 38.8 | 13.5 | 0.153 | 0.019 | 33.2 |

Table 4: **BlendedMVS experiment.** For each scene, we present the number of input images (denoted $N_c$) and the fraction of outliers. For each model, we show the number of images used for reconstruction (denoted $N_r$) and the mean values of the rotation error (in degrees), translation error, and runtime (in seconds). The best results are marked in bold and the second best are underlined.

| Scene | $N_c$ | Out.% | Ours $N_r$ | Rot | Trans | Time | MASt3R $N_r$ | Rot | Trans | Time | VGGSfM $N_r$ | Rot | Trans | Time | Theia $N_r$ | Rot | Trans | Time | COLMAP $N_r$ | Rot | Trans | Time | GLOMAP $N_r$ | Rot | Trans | Time |
|---|---|---|---|---|---|---|---|---|---|---|---|---|---|---|---|---|---|---|---|---|---|---|---|---|---|---|
| scene0 | 75 | 2.0 | 75 | 0.016 | 0.0007 | 54 | 75 | 0.501 | 0.191 | 516 | 75 | 0.045 | 0.0106 | 61 | 75 | 0.009 | 0.0017 | 49 | 75 | 0.006 | 0.0005 | 106 | 75 | 0.007 | 0.0017 | 198 |
| scene1 | 51 | 1.4 | 51 | 0.011 | 0.0021 | 32 | 51 | 0.919 | 0.173 | 1017 | 51 | 0.098 | 0.0112 | 32 | 51 | 0.029 | 0.0099 | 18 | 51 | 0.007 | 0.0003 | 67 | 51 | 0.024 | 0.0102 | 117 |
| scene2 | 33 | 2.2 | 33 | 0.009 | 0.0006 | 21 | 33 | 1.972 | 0.130 | 117 | 33 | 0.227 | 0.0180 | 30 | 33 | 0.045 | 0.0098 | 15 | 33 | 0.003 | 0.0002 | 55 | 33 | 0.025 | 0.0060 | 87 |
| scene3 | 66 | 8.8 | 66 | 0.007 | 0.0007 | 52 | 66 | 0.927 | 0.045 | 815 | 66 | 0.372 | 0.0174 | 52 | 66 | 0.019 | 0.0018 | 21 | 66 | 0.004 | 0.0002 | 128 | 66 | 0.008 | 0.0017 | 392 |

the fine-tuning phase is the most time-consuming part of our pipeline. Table 6 further compares the utilization of resources of the deep-based methods. Our method utilizes slightly more parameters than (Moran et al., 2021) and is two orders of magnitude smaller than (Brynte et al., 2023). In terms of memory usage and proceeding speed our method is the most efficient.

**Classification performance.** Table 7 (left) presents various classification metrics to assess the performance of our inlier/outlier classification module. After removing the predicted outliers, our method significantly decreases the percentage of outliers in the point tracks. These classification results enable the robust BA module to perform well and produce accurate reconstruction and camera pose estimation, yielding another significant decrease in the outlier ratio. Removing these remaining outliers is challenging since these outlier matches survived the RANSAC preprocessing.

## 4.5 ABLATIONS

For ablations, we first examine the impact of our permutation sets-of-sets equivariant architecture. We replace our inlier/outlier classifier module with a set equivariant PointNet architecture. The input in this experiment consists of a set of quadruples, $(x, y, c, t)$, where $(x, y)$ are the keypoint coordinates, $c$ represents camera id, and $t$ denotes track id. As shown in Table 7, our equivariant architecture achieves superior classification accuracies, justifying the importance of using sets-of-sets equivariance.

Next, we examine the importance of the equivariant features. We compare our method, which is trained end-to-end, to (1) the case that only the classification head is trained while the feature encoder is frozen (i.e., trained as in ESFM without outlier classification), and (2) the same architecture but

Table 5: **Runtime.** Given the same contaminated point tracks, we compare the runtime of our proposed method to classical methods, including COLMAP, Theia, and GLOMAP.

| Scene | $N_c$ | Outliers% | Ours | | | | | | COLMAP | | | Theia | | | GLOMAP | | |
|---|---|---|---|---|---|---|---|---|---|---|---|---|---|---|---|---|---|
| | | | Inference (Secs) | Fine-tuning (Secs) | BA (Secs) | Total (Mins) | $N_r$ | $N_r/t$ ↑ | Total (Mins) | $N_r$ | $N_r/t$ ↑ | Total (Mins) | $N_r$ | $N_r/t$ ↑ | Total (Mins) | $N_r$ | $N_r/t$ ↑ |
| Alamo | 573 | 32.6 | 0.004 | 674.3 | 355.6 | 17.2 | 484 | 28.2 | 83.7 | 568 | 6.8 | 13.4 | 553 | **41.4** | 40.0 | 557 | 13.9 |
| Ellis Island | 227 | 25.1 | 0.004 | 103.4 | 66.1 | 2.8 | 214 | 75.9 | 14.9 | 223 | 15.0 | 1.1 | 213 | **201.8** | 7.7 | 219 | 28.6 |
| Madrid Metropolis | 333 | 39.4 | 0.003 | 286.8 | 61.2 | 5.8 | 244 | 42.1 | 25.1 | 323 | 12.9 | - | - | - | 7.1 | 320 | **45.2** |
| Montreal Notre Dame | 448 | 31.7 | 0.005 | 190.62 | 174.9 | 6.1 | 346 | 56.7 | 35.9 | 447 | 12.5 | 3.7 | 422 | **114.6** | 13.5 | 444 | 32.9 |
| Notre Dame | 549 | 35.6 | 0.003 | 1101.75 | 229.4 | 22.2 | 517 | 23.3 | 72.6 | 546 | 7.5 | 11.6 | 534 | **46.0** | 21.1 | 543 | 25.8 |
| NYC Library | 330 | 33.6 | 0.004 | 153.15 | 88.0 | 4.0 | 224 | 55.7 | 26.6 | 330 | 12.4 | 1.5 | 314 | **204.2** | 7.3 | 323 | 44.5 |
| Piazza del Popolo | 336 | 33.1 | 0.002 | 91.6 | 70.0 | 2.7 | 249 | 92.6 | 9.6 | 334 | 34.9 | 3.0 | 325 | **108.8** | 5.9 | 331 | 56.0 |
| Tower of London | 467 | 27.0 | 0.003 | 271.65 | 84.0 | 5.9 | 94 | 15.9 | 65.0 | 467 | 7.2 | 3.1 | 448 | **142.5** | 23.5 | 466 | 19.8 |
| Vienna Cathedral | 824 | 31.4 | 0.005 | 470.35 | 963.5 | 23.9 | 479 | 20.0 | 98.9 | 824 | 8.3 | 11.2 | 772 | **68.8** | 41.6 | 822 | 19.8 |
| Yorkminster | 432 | 29.0 | 0.002 | 270 | 192.5 | 7.7 | 331 | 42.9 | 31.4 | 419 | 13.3 | 2.9 | 390 | **135.3** | 14.8 | 418 | 28.2 |

Table 6: **Resources.** Performance comparison between different methods in terms of a number of parameters, maximum memory usage, and processing speed (images per minute) averaged over the 1dsfm scenes.

| Method | Ours | ESFM | GASFM |
|---|---|---|---|
| #Params (millions) ↓ | 0.73 | **0.66** | 145.17 |
| Max Memory (GBs) ↓ | **9.5** | 10.1 | 25.7 |
| Image/minutes ↑ | **45.3** | 40.9 | 10.7 |

with the classification head removed (equivalent ESFM). It can be seen in Table 8 that training the feature encoder in an end-to-end schedule makes a crucial impact on the accuracy of our method.

Finally, in Appendix D, we validate the importance of our Robust BA; we compare it to a standard BA, showing that the accuracy is significantly improved with our Robust BA.

Table 7: **Classification metric and architecture ablation.** Inlier/outlier classification accuracies and the fraction of outliers predicted with our permutation-equivariant network (sets of sets) compared to an alternative set network (PointNet architecture).

| Scene | $N_c$ | Outliers% Input | Ours | | | | | | PointNet architecture | | | | |
|---|---|---|---|---|---|---|---|---|---|---|---|---|---|
| | | | Recall (inliers) | Recall (outliers) | Precision (outliers) | F-score (outliers) | Outliers% Predicted ↓ | Outliers% Robust BA ↓ | Recall (inliers) | Recall (outliers) | Precision (outliers) | F-score (outliers) | Outliers% Predicted ↓ |
| Alamo | 573 | 32.6 | 65.8 | 74.3 | 51.5 | 60.8 | 16.0 | 10.8 | 36.4 | 74.3 | 36.3 | 48.8 | 25.7 |
| Ellis Island | 227 | 25.1 | 61.9 | 70.4 | 38.4 | 49.7 | 13.9 | 5.7 | 31.1 | 70.8 | 25.8 | 37.8 | 24.0 |
| Madrid Metropolis | 333 | 39.4 | 62.6 | 62.3 | 51.7 | 56.5 | 27.9 | 30.0 | 36.2 | 71.6 | 41.9 | 52.9 | 33.6 |
| Montreal Notre Dame | 448 | 31.7 | 61.5 | 70.4 | 46.4 | 56.0 | 18.6 | 10.8 | 33.8 | 71.6 | 33.9 | 46.0 | 28.5 |
| Notre Dame | 549 | 35.6 | 62.8 | 61.4 | 48.5 | 54.2 | 26.0 | 17.7 | 33.8 | 71.0 | 35.7 | 47.5 | 30.8 |
| NYC Library | 330 | 33.6 | 51.7 | 81.1 | 46.5 | 59.1 | 15.9 | 9.2 | 33.7 | 68.0 | 37.0 | 47.9 | 35.2 |
| Piazza del Popolo | 336 | 33.1 | 56.7 | 78.1 | 47.5 | 59.0 | 16.3 | 7.5 | 26.1 | 81.1 | 35.5 | 49.4 | 26.6 |
| Tower of London | 467 | 27.0 | 28.3 | 85.4 | 30.3 | 44.8 | 15.9 | 2.9 | 42.5 | 59.0 | 27.3 | 37.3 | 26.1 |
| Vienna Cathedral | 824 | 31.4 | 58.5 | 65.8 | 42.2 | 51.5 | 21.2 | 13.6 | 38.3 | 64.9 | 32.6 | 43.4 | 29.7 |
| Yorkminster | 432 | 29.0 | 50.6 | 71.8 | 37.5 | 49.3 | 18.7 | 10.2 | 48.7 | 53 | 29.9 | 38.2 | 28.5 |
| Mean | 451.9 | 31.9 | 56.0 | 72.1 | 44.1 | 54.1 | 19.0 | 11.8 | 36.1 | 68.5 | 33.6 | 44.9 | 28.9 |

Table 8: **Encoder ablation.** For each scene, we show the number of input images (denoted $N_c$) and the fraction of outliers. For each model, we show the number of images used for reconstruction (denoted $N_r$) and the mean values of the rotation (in degrees) and translation errors.

| Scene | $N_c$ | Outliers% | Ours | | | Frozen Encoder | | | No Encoder (ESFM) | | |
|---|---|---|---|---|---|---|---|---|---|---|---|
| | | | $N_r$ | Rot | Trans | $N_r$ | Rot | Trans | $N_r$ | Rot | Trans |
| Alamo | 573 | 32.6 | 484 | **3.66** | 0.515 | 511 | 3.79 | 0.394 | 457 | 4.53 | **0.338** |
| Ellis Island | 227 | 25.1 | **214** | **0.82** | **0.122** | 198 | 20.60 | 2.094 | 196 | 21.13 | 2.091 |
| Madrid Metropolis | 333 | 39.4 | **244** | **8.42** | 0.827 | 231 | 9.09 | **0.794** | 151 | 19.56 | 1.932 |
| Montreal Notre Dame | 448 | 31.7 | **346** | 2.82 | 0.352 | 335 | **1.17** | **0.244** | 309 | 10.13 | 1.778 |
| Notre Dame | 549 | 35.6 | 517 | 1.2 | 0.231 | **518** | **0.69** | **0.168** | 487 | 1.95 | 0.233 |
| NYC Library | 330 | 33.6 | 224 | **3.96** | **0.429** | **232** | 4.07 | 0.542 | 177 | 4.42 | 0.546 |
| Piazza del Popolo | 336 | 33.1 | 249 | **2.20** | **0.186** | **254** | 5.59 | 0.739 | 204 | 10.77 | 1.006 |
| Tower of London | 467 | 27.0 | 94 | **0.67** | **0.026** | 141 | 25.93 | 1.601 | **196** | 22.02 | 2.226 |
| Vienna Cathedral | 824 | 31.4 | 479 | **1.52** | **0.112** | 523 | 2.94 | 0.222 | **551** | 6.52 | 0.553 |
| Yorkminster | 432 | 29.0 | 331 | 14.54 | 1.468 | **354** | 13.24 | 1.183 | 215 | **10.63** | **0.636** |

## 5 CONCLUSION

We present a permutation equivariant architecture for robust multiview structure from motion. By integrating a sets-of-sets equivariant inlier/outlier classification module, our proposed method copes well with point-track tensors contaminated with many outliers originating from scenes that include hundreds of images. In addition, we modified the bundle adjustment module to make it robust enough to handle classification errors. Our method successfully handles challenging datasets that include hundreds of uncontrolled internet images, achieving highly accurate recovery, superior to existing deep methods and on par with state-of-the-art classical methods. However, we observed cases in which our method uses only a subset of the input cameras. This occurs due to an excess removal of predicted outliers, which might yield an unconnected viewing graph. We plan to address this limitation in our future work.

ACKNOWLEDGMENTS

This research was supported in part by the Israel Science Foundation, grant No. 1639/19, by the Israeli Council for Higher Education (CHE) via the Weizmann Data Science Research Center, by the MBZUAI-WIS Joint Program for Artificial Intelligence Research and by research grants from the Estates of Bernice Bernath and Marni Josephs Grossman; Joel B. Levey; Tully and Michele Plesser and the Anita James Rosen and Harry Schutzman Foundations.

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

APPENDIX

## A    QUALITATIVE RESULTS

Figures 3 and 4 show reconstruction examples with our method, compared to other deep-based methods.

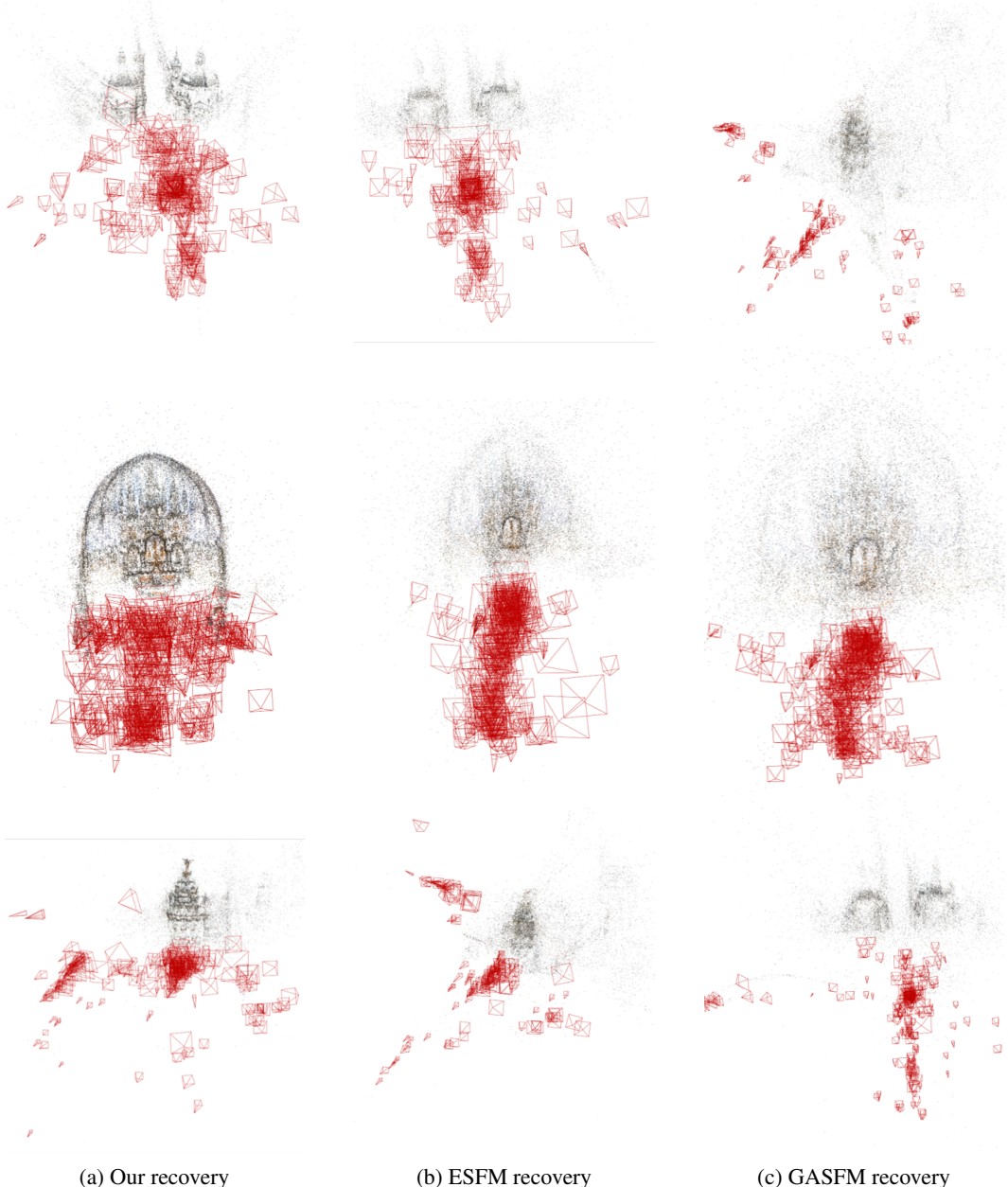

    (a) Our recovery                 (b) ESFM recovery                 (c) GASFM recovery

Figure 3: **Reconstruction results on scenes from the 1DSFM dataset.** The figure shows 3D reconstructions and camera pose estimation. The triplet in each row shows reconstruction with our method (left), ESFM (Moran et al., 2021) (middle), and GASFM (Brynte et al., 2023) (right). The scenes are Piazza del Popolo (top row, 33.1% outliers), Montreal Notre Dame (middle row, 33% outliers), and Madrid Metropolis (bottom row, 39.4% outliers).

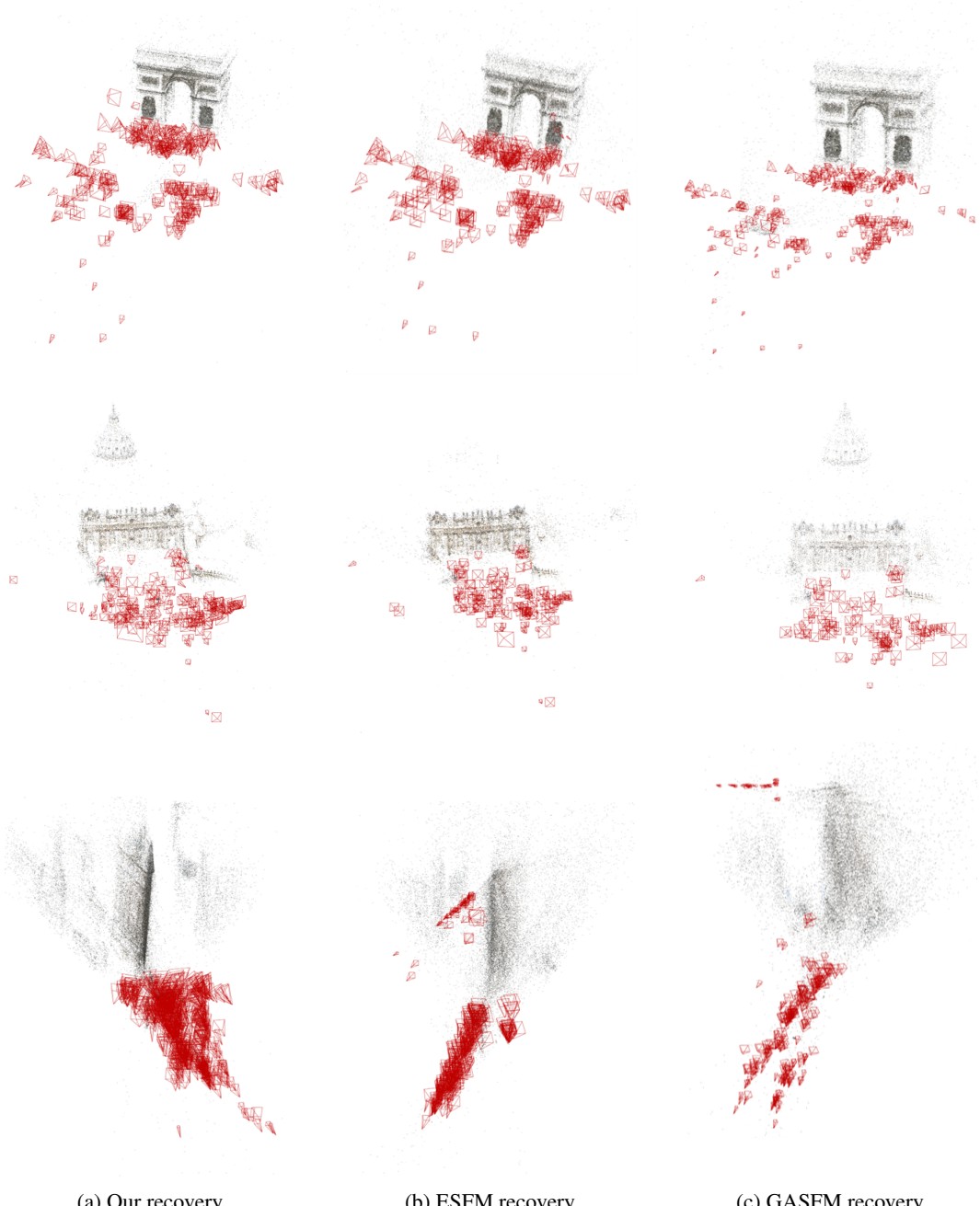

(a) Our recovery        (b) ESFM recovery        (c) GASFM recovery

Figure 4: **Reconstruction results on scenes from MegaDepth dataset.** The figure shows 3D reconstructions and camera pose estimation. The triplet in each row shows reconstruction with our method (left), ESFM (Moran et al., 2021) (middle), and GASFM (Brynte et al., 2023) (right). The scenes are 0012 (top row, 16.3% outliers), 0015 (middle row, 20.6% outliers), and 0060 (bottom row, 41.6% outliers).

# B  IMPLEMENTATION DETAILS

**Our code and preprocessed point tracks data will be made publicly available.**

**Framework.**    Our method was trained and evaluated on NVIDIA A40 GPUs (48GB of GPU Memory). We used PyTorch (Paszke et al., 2019) as the deep learning framework and the ADAM optimizer (Kingma & Ba, 2014) with normalized gradients.

**Training.** During training, for each epoch, we processed all training scenes sequentially. For each scene, we randomly selected a subset of 10%-20% of the images in the scene. We used a validation set for early stopping. The validation and test sets were evaluated by using the complete point tracks matrix. The training time of our model on the Megadepth dataset took roughly 16 hours on a single NVIDIA A40 GPU. We fix the seed to 20.

**Architecture details.**  We use normalized point tracks $\mathbf{x}_{ij}$, as inputs to our method which are normalized using the known intrinsic parameters. The shared features encoder $E$ has 3 layers, each with 256 feature channels and ReLU activation. The camera head $H_{\mathrm{cams}}$, 3D point head $H_{\mathrm{pts}}$, and the outliers classification head $H_{\mathrm{outlier}}$, each have 3 layers with 256 channels. After each layer in $E$ we normalize its features by subtracting their mean.

**Hyper-parameter search.** We tried different implementation hyper-parameters including (1) learning rates $\in \{1e-2, 1e-3, 1e-4\}$, (2) network width $\in \{128, 256, 512\}$ for the encoder $E$ and the heads, (3) number of layers $\in \{2, 3, 4, 5\}$ in these networks, and (4) threshold for outlier removal $\in \{0.4, 0.5, 0.6, 0.7, 0.8\}$. For the Strecha and BlendedMVS datasets, we chose a threshold of 0.8, as these datasets are captured with a single camera and therefore contain fewer outliers.

**Bundle adjustment** For the bundle adjustment, we employed the Ceres Solver's implementation of bundle adjustment Agarwal et al. with the Huber loss to enhance robustness (with parameter 0.1). In each bundle adjustment round, we limited the number of iterations to 300 or until convergence, whichever occurred first.

# C  CONSTRUCTING POINT TRACKS

Our preprocessing step for constructing the point tracks is as follows:

1. Extracting SIFT (Lowe, 2004) features for all the images.
2. Applying exhaustive pair-wise RANSAC (Fischler & Bolles, 1981) matching for all image pairs.
3. Chaining each sequence of two-view matches for creating a point track.

We discard a 3D point and its track if it is viewed in less than 3 cameras or if it includes an inconsistent cycle, i.e., we detect two key points in the same image for the same point track.

The above preprocess defines a valid point track structure for our architecture. We next define the inliers/outliers labeling process, which assumes that every training scene has an associated COLMAP (Schönberger & Frahm, 2016) reconstruction:

1. For each track within our point track set:
    (a) Identify the corresponding track in COLMAP's reconstruction output
    (b) For every keypoint in the track:
        - Label the keypoint as an outlier if it does not appear in the corresponding COLMAP track; otherwise, label it as an inlier.
2. Remove keypoints labeled as outliers from the point tracks.
3. Triangulate the 3D points from the cleaned point tracks using the ground truth camera poses obtained from COLMAP.
4. Calculate the reprojection error for the triangulated 3D points across the entire point tracks, including the ones labeled initially as outliers.
5. Assign an inlier label to each keypoint whose reprojection error is below 4 pixels; keypoints exceeding this threshold are classified as outliers.

# D ADDITIONAL RESULTS

Here we show median results for the MegaDepth and 1DSFM experiments (Tables 9 and 10). We further validate the importance of our Robust BA compared to a standard BA (Table 11).

Table 9: MegaDepth experiment. The table shows the **median** values of the rotation (in degrees) and translation errors. (Above the middle rule are Group 1 scenes with <1000 images; below are Group 2 scenes with >1000 images, subsampled to 300 for testing.) Winning results are marked in bold and underlined. Yellow represents the best result among the deep-based algorithms, and green among the classical algorithms.

| Scene | $N_c$ | Outliers% | **Ours** | | | ESFM | | | GASFM | | | Theia | | | GLOMAP | | | ESFM* | | |
|---|---|---|---|---|---|---|---|---|---|---|---|---|---|---|---|---|---|---|---|---|
| | | | $N_r$ | Rot | Trans | $N_r$ | Rot | Trans | $N_r$ | Rot | Trans | $N_r$ | Rot | Trans | $N_r$ | Rot | Trans | $N_r$ | Rot | Trans |
| 0238 | 522 | 44.6 | 283 | 0.72 | 0.043 | 76 | 1.75 | 0.100 | 76 | 5.15 | 0.323 | 506 | 0.54 | 0.109 | 499 | 0.22 | 0.043 | 512 | 0.64 | 0.038 |
| 0060 | 528 | 41.6 | 503 | 0.14 | 0.011 | 303 | 8.54 | 1.268 | 258 | 13.81 | 2.744 | 525 | 0.26 | 0.039 | 522 | 0.04 | 0.012 | 524 | 0.02 | 0.003 |
| 0197 | 870 | 40.7 | 667 | 2.06 | 0.133 | 281 | 4.14 | 0.183 | 454 | 6.65 | 0.688 | 855 | 0.77 | 0.118 | 814 | 0.13 | 0.016 | 825 | 0.11 | 0.008 |
| 0094 | 763 | 40.1 | 537 | 0.38 | 0.015 | 93 | 17.62 | 2.048 | 359 | 0.88 | 0.028 | 742 | 0.21 | 0.033 | 717 | 0.20 | 1.957 | 643 | 0.18 | 0.014 |
| 0265 | 571 | 38.8 | 346 | 0.74 | 0.209 | 270 | 14.64 | 1.666 | 274 | 18.91 | 1.716 | 554 | 4.11 | 1.651 | 558 | 6.66 | 1.889 | 559 | 0.06 | 0.022 |
| 0083 | 635 | 31.3 | 596 | 0.15 | 0.009 | 568 | 0.90 | 0.027 | 574 | 2.48 | 0.099 | 632 | 0.15 | 0.013 | 614 | 0.04 | 0.007 | 556 | 16.03 | 0.973 |
| 0076 | 558 | 30.5 | 524 | 0.11 | 0.010 | 454 | 1.60 | 0.091 | 431 | 0.05 | 0.006 | 549 | 0.44 | 0.058 | 541 | 0.08 | 0.017 | 547 | 0.03 | 0.004 |
| 0185 | 368 | 30.0 | 350 | 0.04 | 0.006 | 261 | 0.53 | 0.037 | 254 | 0.46 | 0.048 | 365 | 0.31 | 0.037 | 365 | 0.11 | 0.012 | 130 | 29.39 | 2.160 |
| 0048 | 512 | 24.2 | 474 | 2.16 | 0.098 | 469 | 1.57 | 0.080 | 481 | 0.60 | 0.028 | 507 | 0.21 | 0.020 | 506 | 0.06 | 0.007 | 508 | 0.05 | 0.002 |
| 0024 | 356 | 23.0 | 309 | 0.58 | 0.046 | 271 | 4.64 | 0.187 | 300 | 0.48 | 0.104 | 355 | 0.24 | 0.091 | 339 | 0.07 | 0.045 | 343 | 1.41 | 0.078 |
| 0223 | 214 | 17.0 | 204 | 1.56 | 0.078 | 191 | 1.81 | 0.180 | 194 | 1.53 | 1.106 | 212 | 0.89 | 0.152 | 214 | 0.41 | 0.046 | 211 | 0.06 | 0.010 |
| 5016 | 28 | 0.2 | 28 | 0.10 | 0.005 | 28 | 0.07 | 0.004 | 28 | 0.06 | 0.003 | 28 | 0.07 | 0.019 | 28 | 0.04 | 0.016 | 28 | 0.02 | 0.003 |
| 0046 | 440 | 14.6 | 399 | 0.78 | 0.028 | 97 | 0.58 | 0.017 | 426 | 0.35 | 0.011 | 434 | 0.16 | 0.016 | 440 | 0.02 | 0.002 | 33 | 29.18 | 1.331 |
| 0099 | 299 | 47.4 | 190 | 3.18 | 0.697 | 104 | 2.21 | 0.402 | 128 | 3.95 | 0.409 | 297 | 1.79 | 0.394 | 255 | 0.06 | 0.026 | 243 | 2.97 | 0.618 |
| 1001 | 285 | 43.9 | 251 | 1.41 | 0.276 | 241 | 2.44 | 0.860 | 261 | 2.26 | 0.386 | 276 | 4.85 | 2.893 | 281 | 3.29 | 2.645 | 280 | 0.08 | 0.053 |
| 0231 | 296 | 42.2 | 246 | 0.38 | 0.014 | 214 | 0.28 | 0.011 | 209 | 0.19 | 0.009 | 286 | 0.58 | 0.072 | 279 | 0.20 | 0.021 | 284 | 0.26 | 0.011 |
| 0411 | 299 | 29.9 | 273 | 0.07 | 0.009 | 188 | 6.21 | 0.338 | 232 | 1.28 | 0.051 | 293 | 0.19 | 0.079 | 269 | 0.09 | 0.036 | 288 | 0.05 | 0.007 |
| 0377 | 295 | 27.5 | 210 | 0.28 | 0.014 | 162 | 0.25 | 0.010 | 167 | 0.09 | 0.008 | 269 | 0.29 | 0.075 | 268 | 0.23 | 0.021 | 279 | 0.52 | 0.022 |
| 0102 | 299 | 25.8 | 284 | 0.07 | 0.007 | 255 | 0.50 | 0.032 | 278 | 0.37 | 0.023 | 294 | 1.03 | 0.114 | 293 | 0.04 | 0.013 | 155 | 13.56 | 2.859 |
| 0147 | 298 | 24.6 | 207 | 2.07 | 0.088 | 197 | 2.05 | 0.097 | 225 | 5.26 | 0.296 | 284 | 1.10 | 0.064 | 290 | 1.78 | 2.056 | 251 | 1.14 | 0.041 |
| 0148 | 287 | 24.6 | 197 | 0.54 | 0.024 | 206 | 0.59 | 0.028 | 209 | 0.59 | 0.028 | 275 | 3.01 | 0.301 | 283 | 3.09 | 1.301 | 249 | 0.46 | 0.023 |
| 0446 | 298 | 22.1 | 288 | 0.41 | 0.013 | 283 | 0.70 | 0.021 | 291 | 0.20 | 0.008 | 289 | 0.61 | 0.073 | 296 | 0.14 | 0.020 | 294 | 0.22 | 0.006 |
| 0022 | 297 | 21.2 | 274 | 0.13 | 0.011 | 241 | 0.51 | 0.025 | 263 | 0.16 | 0.013 | 296 | 0.28 | 0.065 | 281 | 0.08 | 0.023 | 289 | 0.08 | 0.008 |
| 0327 | 298 | 21.0 | 271 | 0.11 | 0.006 | 281 | 0.75 | 0.022 | 284 | 0.10 | 0.007 | 288 | 0.73 | 0.087 | 299 | 7.14 | 0.333 | 294 | 0.03 | 0.003 |
| 0015 | 284 | 20.6 | 215 | 0.27 | 0.021 | 142 | 1.30 | 0.105 | 149 | 6.35 | 0.556 | 244 | 0.42 | 0.084 | 274 | 0.11 | 0.014 | 185 | 0.43 | 0.053 |
| 0455 | 298 | 19.8 | 293 | 0.18 | 0.010 | 293 | 0.23 | 0.014 | 294 | 0.29 | 0.016 | 294 | 0.36 | 0.047 | 298 | 0.14 | 0.017 | 298 | 0.26 | 0.012 |
| 0496 | 297 | 19.2 | 281 | 0.13 | 0.006 | 281 | 2.81 | 0.197 | 277 | 0.44 | 0.027 | 285 | 0.61 | 0.080 | 291 | 0.16 | 0.028 | 293 | 0.23 | 0.009 |
| 1589 | 299 | 17.4 | 290 | 0.08 | 0.003 | 284 | 0.40 | 0.008 | 283 | 1.00 | 0.016 | 288 | 0.32 | 0.057 | 299 | 0.03 | 0.007 | 296 | 0.20 | 0.004 |
| 0012 | 299 | 16.3 | 287 | 0.39 | 0.023 | 291 | 2.04 | 0.131 | 294 | 0.21 | 0.018 | 129 | 0.56 | 0.092 | 295 | 0.20 | 0.017 | 294 | 0.22 | 0.013 |
| 0104 | 284 | 16.2 | 193 | 0.16 | 0.009 | 220 | 11.31 | 0.568 | 228 | 1.32 | 0.051 | 265 | 9.24 | 0.658 | 280 | 9.86 | 0.550 | 200 | 0.49 | 0.019 |
| 0019 | 299 | 15.4 | 250 | 0.04 | 0.004 | 267 | 2.46 | 0.128 | 293 | 0.58 | 0.023 | 271 | 0.31 | 0.030 | 296 | 0.04 | 0.004 | 296 | 2.23 | 0.097 |
| 0063 | 293 | 14.5 | 262 | 0.26 | 0.013 | 262 | 1.18 | 0.057 | 257 | 0.40 | 0.020 | 268 | 0.45 | 0.063 | 288 | 0.17 | 0.017 | 275 | 0.15 | 0.009 |
| 0130 | 285 | 14.4 | 192 | 0.10 | 0.005 | 192 | 0.94 | 0.030 | 194 | 1.00 | 0.033 | 187 | 0.63 | 0.072 | 281 | 0.94 | 0.535 | 282 | 0.82 | 0.041 |
| 0080 | 284 | 12.9 | 139 | 0.27 | 0.010 | 137 | 0.32 | 0.012 | 139 | 1.36 | 0.054 | 278 | 1.84 | 0.335 | 283 | 1.71 | 0.169 | 163 | 0.32 | 0.011 |
| 0240 | 298 | 11.9 | 275 | 1.56 | 0.090 | 274 | 1.14 | 0.064 | 272 | 2.00 | 0.113 | 278 | 0.47 | 0.057 | 294 | 0.17 | 0.041 | 296 | 0.09 | 0.006 |
| 0007 | 290 | 11.7 | 172 | 0.23 | 0.010 | 260 | 25.50 | 1.432 | 280 | 0.18 | 0.010 | 277 | 0.69 | 0.071 | 290 | 0.06 | 0.006 | 286 | 0.96 | 0.053 |
| Mean | 379 | 25.6 | 298 | 0.61 | 0.057 | 239 | 3.46 | 0.291 | 267 | 2.25 | 0.252 | 346 | 1.08 | 0.228 | 353 | 1.05 | 0.332 | 319 | 2.86 | 0.240 |

Table 10: 1DSFM experiment. The table shows the **median** values of the rotation (in degrees), and translation errors. Winning results are marked in bold and underlined. Yellow represents the best result among the deep-based algorithms and green among the classical algorithms.

| Scene | $N_c$ | Outliers% | **Ours** | | | ESFM | | | GASFM | | | Theia | | | GLOMAP | | | ESFM* | | |
|---|---|---|---|---|---|---|---|---|---|---|---|---|---|---|---|---|---|---|---|---|
| | | | $N_r$ | Rot | Trans | $N_r$ | Rot | Trans | $N_r$ | Rot | Trans | $N_r$ | Rot | Trans | $N_r$ | Rot | Trans | $N_r$ | Rot | Trans |
| Alamo | 573 | 32.6 | 484 | 0.97 | 0.037 | 457 | 1.00 | 0.047 | 448 | 2.64 | 0.115 | 553 | 2.29 | 0.539 | 557 | 0.61 | 0.144 | 526 | 0.35 | 0.016 |
| Ellis Island | 227 | 25.1 | 214 | 0.32 | 0.036 | 196 | 18.88 | 1.554 | 198 | 19.24 | 1.548 | 213 | 3.85 | 0.712 | 219 | 0.46 | 0.087 | 220 | 0.19 | 0.026 |
| Madrid Metropolis | 333 | 39.4 | 244 | 4.42 | 0.193 | 151 | 19.85 | 1.641 | 159 | 21.76 | 1.560 | - | - | - | 320 | 0.53 | 0.096 | 290 | 23.71 | 2.154 |
| Montreal Notre Dame | 448 | 31.7 | 346 | 1.00 | 0.056 | 309 | 4.42 | 0.790 | 311 | 4.83 | 0.945 | 422 | 2.63 | 0.808 | 444 | 0.40 | 0.158 | 414 | 0.07 | 0.009 |
| Notre Dame | 549 | 35.6 | 517 | 0.55 | 0.025 | 487 | 0.43 | 0.025 | 499 | 0.51 | 0.025 | 314 | 1.54 | 0.133 | 543 | 1.15 | 0.130 | 528 | 0.30 | 0.012 |
| NYC Library | 330 | 33.6 | 224 | 1.48 | 0.074 | 177 | 1.93 | 0.102 | 218 | 3.74 | 0.264 | 534 | 1.65 | 0.360 | 323 | 0.46 | 0.075 | 301 | 1.16 | 0.087 |
| Piazza del Popolo | 336 | 33.1 | 249 | 0.80 | 0.034 | 204 | 3.20 | 0.353 | 198 | 3.79 | 0.583 | 325 | 1.15 | 0.342 | 331 | 0.28 | 0.084 | 303 | 0.88 | 0.052 |
| Tower of London | 467 | 27.0 | 94 | 0.48 | 0.012 | 196 | 13.59 | 0.564 | 152 | 33.19 | 2.339 | 448 | 3.23 | 0.527 | 466 | 0.42 | 0.071 | 213 | 4.53 | 0.163 |
| Vienna Cathedral | 824 | 31.4 | 479 | 0.48 | 0.016 | 551 | 1.40 | 0.046 | 558 | 1.67 | 0.044 | 772 | 9.32 | 0.838 | 822 | 0.61 | 0.206 | 536 | 0.44 | 0.013 |
| Yorkminster | 432 | 29.0 | 331 | 4.67 | 0.299 | 215 | 6.15 | 0.302 | 251 | 7.80 | 0.378 | 390 | 4.26 | 0.948 | 418 | 0.60 | 0.069 | 389 | 6.48 | 0.265 |

Table 11: **Bundle-Adjustment (BA) ablation.** We compare the effect of post-processing with a robust vs. standard BA. For each scene, we show the number of input images (denoted $N_c$) and the fraction of outliers. For each model, we show the number of images used for reconstruction (denoted $N_r$) and the mean values of the rotation (in degrees) and translation errors.

| Scene | $N_c$ | Outliers% | Robust BA (Ours) | | | Standard BA) | | |
|---|---|---|---|---|---|---|---|---|
| | | | $N_r$ | Rot | Trans | $N_r$ | Rot | Trans |
| Alamo | 573 | 32.6 | 484 | **3.66** | **0.515** | **568** | 10.18 | 1.970 |
| Ellis Island | 227 | 25.1 | 214 | **0.82** | **0.122** | **227** | 4.05 | 1.045 |
| Madrid Metropolis | 333 | 39.4 | 244 | **8.42** | **0.827** | **333** | 20.07 | 2.352 |
| Montreal Notre Dame | 448 | 31.7 | 346 | **2.82** | **0.352** | **447** | 4.52 | 0.563 |
| Notre Dame | 549 | 35.6 | 517 | **1.20** | **0.231** | **548** | 3.45 | 0.378 |
| NYC Library | 330 | 33.6 | 224 | **3.96** | **0.429** | **329** | 24.94 | 3.571 |
| Piazza del Popolo | 336 | 33.1 | 249 | **2.20** | **0.186** | **335** | 21.68 | 2.001 |
| Tower of London | 467 | 27.0 | 94 | **0.67** | **0.026** | **465** | 57.41 | 3.347 |
| Vienna Cathedral | 824 | 31.4 | 479 | **1.52** | **0.112** | **819** | 43.38 | 2.784 |
| Yorkminster | 432 | 29.0 | 331 | **14.54** | **1.468** | **431** | 21.17 | 3.660 |

