# OpenReview forum: "RESfM: Robust Deep Equivariant Structure from Motion"
_ICLR.cc/2025/Conference — ICLR 2025 Poster_

### Official Review · Reviewer_oVEX · 2024-10-16

**Soundness:** 2
**Presentation:** 2
**Contribution:** 1
**Rating:** 5
**Confidence:** 4

**Summary:**

This paper studies an important problem in multi-view structure from motion, specifically aiming to recover 3D points and camera poses from input point tracks. To this end, based on an existing permutation equivariant architecture, an outlier classification module is integrated to cope with outliers. A robust Bundle Adjustment method is also proposed, based on recursive potential outlier removal. Experiments are conducted on multiple datasets, demonstrating the robustness of the proposed method against existing ones.

**Strengths:**

The paper is well-presented in general and easy to understand, with a comprehensive literature review and extensive experiments.

**Weaknesses:**

The paper studies an interesting and practical problem, but the main weakness is its limited contribution.

1. The proposed method is largely built on existing network architecture, the difference being only that outlier classification output channels are added.

2. The proposed solution for handling outliers incurs much additional overhead in the recursive "finetune" process, lacks theoretical justification, comprises inliers recall, and can not reject outliers well (according to Table 7). Thus the contribution of the method is limited. The same problem also goes for the proposed robust Bundle Adjustment method.

**Questions:**

Experiment-wise, why ESFM* has much larger rotation error in some cases, and why GLOMAP has good rotation predictions but not translation predictions. And it might be better to give the mean errors in the table for more straightforward comparisons.

---

> ### Author Response · Authors · 2024-11-30
>
> We thank the reviewer for his thoughtful feedback.
>
> **W1. Contribution**
>
>  For further clarification on the significance and novelty of our method, we kindly refer you to our general response and the detailed reply to reviewer PnnE. We hope this provides a clearer understanding of the contributions of our work.
>
> **W2.**
>
> We are not sure we fully understand your concern. Our method first classifies each keypoint as an inlier or outlier, then removes the outliers from the tracks as predicted by the classifier. Afterward, we fine-tune the network and apply Bundle Adjustment (BA). This process ensures that outliers are effectively handled and does not result in excessive overhead. Additionally, our method significantly reduces the percentage of outliers. Could you please clarify which part of this process you believe introduces unnecessary overhead or leads to poor outlier rejection?
>
> **Q1. GLOMAP and ESFM\* results**
>
>  Thanks to your insightful question, we discovered a bug in the GLOMAP code that caused a few unregistered cameras to be included in the reconstruction. This issue has now been addressed, and we have updated the results in the revised version of the paper.
> Regarding ESFM*, we believe the larger rotation errors in some cases stem from limitations inherent to the ESFM method itself. However, despite these occasional errors, ESFM* is still able to successfully reconstruct more scenes overall.

---

### Official Review · Reviewer_D44J · 2024-11-04

**Soundness:** 3
**Presentation:** 3
**Contribution:** 3
**Rating:** 8
**Confidence:** 4

**Summary:**

In this work, authors robustify the existing deep equivariant SfM method from [1] by incorporating an inlier-outlier prediction module and robust Bundle Adjustment. Specifically, deep SfM uses projective factorization on a 2D point track matrix, but the method in [1] assumes that this matrix is complete and outlier-free. Authors in this work argue that these assumptions are unrealistic. Hence, authors proposed a deep equivariant SfM that works on a large collection of internet photos and works with a matrix of point tracks that might have outliers. In their method, they first train an end-to-end outlier prediction and camera poses + 3D point prediction network from a matrix of 2D point tracks. At test time, they use "inlier-outlier" predictions from the pre-trained network to remove outliers and fine-tune the network for 3D pose + 3D points recovery using unsupervised losses for each scene.

Authors show impressive results across different datasets. Their method achieves the best results among deep SfM methods and achieves comparable results to the best classical SfM methods.

References:

[1] Moran, Dror, et al. "Deep permutation equivariant structure from motion." Proceedings of the IEEE/CVF International Conference on Computer Vision. 2021.

**Strengths:**

Paper is well-written with a clear objective being robustifying the method proposed in [1]. Overall content of the paper is well-written, coherent, and easy to understand and follow.

The proposed inlier-outlier prediction head shows improvement compared to ESFM [1]. Results across many scenes of indoor and outdoor datasets show that the proposed method improves over [1] and achieves comparable results to the best classical SfM methods.

Authors use unsupervised reprojection losses. This means that the network can be easily fine-tuned for different scenes, removing a limitation of deep SfM networks that they cannot generalize to scenes outside their training sets.

References:

[1] Moran, Dror, et al. "Deep permutation equivariant structure from motion." Proceedings of the IEEE/CVF International Conference on Computer Vision. 2021.

**Weaknesses:**

Authors mention that they did not use Mast3R [2] as one of the baseline method because it does not work with large number of images. But, I do think Mast3R [2] is able to work with large sets of images. At least on Stretcha and BlendedMVS experiments, authors should be able to use Mast3R for full 3D reconstruction

What is not clear from the paper is whether other methods use Robust Bundle Adjustment proposed in this work. I agree robust BA is necessary for accurate reconstruction, but the contribution of the paper is robust "deep equivariant SfM". For apples-to-apples comparison, authors should also compare with Standard BA or authors should apply robust BA post-processing to compared methods. I see that authors do provide ablation study for effectiveness of robust BA, but to verify the claim of deep robust sfm, it would be great to compare all the methods with either regular BA or robust BA.

References:

[2] Leroy, Vincent, Yohann Cabon, and Jérôme Revaud. "Grounding Image Matching in 3D with MASt3R." arXiv preprint arXiv:2406.09756 (2024).

**Questions:**

Can we compare with Mast3R / Spann3R [3]? These methods are new way of doing scene reconstruction and does show promise. They are also deep "reconstruction" methods because they use deep networks directly to get 3D pointmaps.  This comparison with help researchers find limitations or benefits of one method over another.

Is it possible to make results table a bit easy to read? Color + bold letters are not visible clearly. It is hard to find best method for each scene. Also, is there a way to get an average of some of metrics across scenes? (or over whole dataset).

References:

[3] Wang, Hengyi, and Lourdes Agapito. "3d reconstruction with spatial memory." arXiv preprint arXiv:2408.16061 (2024).

---

> ### Author Response · Authors · 2024-11-30
>
> Thank you for your thoughtful feedback.
>
> **W1. Comparison with MASt3R**
>
> We tested MASt3R (Leroy et al., 2024) on all the test scenes. However, it successfully completed runs only on the Strecha and BlendedMVS datasets, as well as Scene 5016 (comprising 28 images); in all other scenes, the runs failed. As detailed in the revised version of the paper (Tables 3 and 4), our method outperforms MASt3R in both rotation and translation accuracy while also being significantly faster.
>
>
> **W2. Robust BA**
>
> Please note that other works also utilize different types of Robust Bundle Adjustment, tailored to their specific method designs. In our case, the proposed Robust Bundle Adjustment is optimized for use with methods like ours, GASFM (Brynte et al., 2023) and ESFM (Moran et al., 2021).
>
>
> **Q1. Comparison with MASt3R**
>
> For a comparison with MASt3R, please refer to our response to W1. Regarding Spann3R [3], while it achieves impressive point cloud reconstructions, it does not estimate camera poses. Thus, a direct comparison with our method, which focuses on camera pose estimation and reconstruction, would not be appropriate.

---

> > ### Comment · Reviewer_D44J · 2024-12-02
> > **Thanks you for a rebuttal!**
> >
> > W1. Comparison with Mast3R
> >
> > Thanks, I am satisfied with the comparison. I didn't expect it to work on large scenes. BlendedMVS and Strecha results look promising.
> >
> > W2. Robust BA
> >
> > I see. So, you used other methods' robust BA as it is?
> >
> > Q1. Comparison
> >
> > Yes, Mast3R comparison is enough. There should be a way to get poses from Spann3R pointcloud, but it is out of scope of this paper.
> >
> >
> > I am satisfied with the response and maintain my rating of 8.

---

> > > ### Author Response · Authors · 2024-12-03
> > >
> > > Thank you for your comments and positive feedback on our work. We appreciate the time you’ve taken to review our paper and are glad to hear that you find the comparisons with Mast3R results promising!
> > >
> > > Regarding your question about robust BA, you are correct that we used the robust BA implementations provided by the other methods without modification. For the ESFM and GASFM methods, which did not originally include robust BA, we applied our own implementation to maintain consistency and ensure a fair evaluation.

---

### Official Review · Reviewer_PnnE · 2024-11-04

**Soundness:** 3
**Presentation:** 3
**Contribution:** 2
**Rating:** 5
**Confidence:** 4

**Summary:**

The paper presents an architecture for Multiview Structure from Motion (SfM) focusing on the robust recovery of camera pose and 3D scene structure from large, uncontrolled image collections that contain numerous outliers. Traditional and deep-based SfM approaches often struggle with outlier point tracks resulting from viewpoint variations, illumination changes, and repetitive structures, significantly impacting their performance. The authors propose an enhancement to the deep network architecture proposed by Moran et al. in 2021, incorporating an outlier classification module within a permutation equivariant framework. The revised architecture also integrates a robust bundle adjustment step to correct classification errors. The method has been tested on challenging datasets like MegaDepth and 1DSFM and has shown good accuracy and competitive runtimes against both contemporary deep-learning methods and established classical techniques.

**Strengths:**

1. This paper improves the existing ESFM framework by integrating a new inlier-outlier classification branch, alongside a robust structure from motion (SfM) mechanism. This approach makes sense, given that outliers represent a primary challenge for SfM methods in real-world applications. Quantitative experiments demonstrate the efficacy of these components.

2. This paper conducts extensive experiments over various datasets to prove their claims.

**Weaknesses:**

1. The main concern about this work is the novelty of the proposed framework. Compared to ESfM, the new designs are just (1) a simple classfication branch to identify inlier/outlier, which conducts simple binary classfication, and (b) robust BA considering high projection error, point track length, and multi-step refinement. Both these two designs have been proven effective over the long time and are not the new techniques from this work. For instance, systems like COLMAP, Theia, and VGGSfM incorporate variants of robust BA. Numerous matching and tracking methodologies, including SuperGlue, LightGlue, PiPs, and Cotracker, implement classification branches to classify inlier/outlier status. While the reviewer acknowledges the effectiveness and importance of these features, there are reservations about their originality as claimed by this work.


2. Moreover, the section of ablation study needs more thinking. The authors discussed "the impact of our permutation sets-of-sets equivariant architecture" and "importance of the equivariant features". However, as far as the reviewer understands, both these two are the benefits from the architecture of ESfM.  To validate the design choices of this paper, the authors should focus on why their error handling designs are better, instead of proving their base model is better.


3. Concerns also arise regarding the reliability of the results. If interpreted correctly, the rotational metrics in the study are reported in degrees, not radians. Notably, in Tables 3 and 4, the reported rotational errors for some methods are as low as 0.02 or 0.03 degrees. The reviewer questions whether such marginal differences are statistically significant enough to discriminate between methods. (e.g., BlendedMVS, despite being half-synthetic, does not offer flawless ground truth)

**Questions:**

The main question is about the Weaknesses 1 and 2, i.e., what's the main novelty of the proposed method compared to the existing literature?

---

> ### Author Response · Authors · 2024-11-30
>
> **W1. Novelty:**
>
> We appreciate the reviewer’s feedback and would like to clarify the novelty of our proposed framework. The main novelty of our work lies in the formulation of the **multi-view outlier removal task**, which has not been addressed before.
>
> In contrast to two-view outlier removal, which primarily focuses on estimating the relative pose between two images and is relatively much simpler, multi-view outlier removal involves integrating multiple images and addressing more complex challenges, such as ensuring consistency across diverse viewpoints. Furthermore, existing tracking methods often assume fixed lighting conditions and a small number of outliers, with the temporal component helping to mitigate errors. In contrast, our approach operates in a global SfM setting, where outlier rates can be much higher, and lighting conditions and camera viewpoints vary significantly.
>
>
> By tackling the multi-view outlier removal problem, we have introduced a new task that opens up avenues for future research. We believe that our large-scale benchmark and proposed framework will serve as a valuable resource for advancing the field and improving performance in this area.
>
> **W2. Ablation:**
>
> Our ablation study, shown in Table 7, focuses on handling errors caused by the presence of outliers in the task of multi-view outlier removal. It demonstrates that the choice of our permutation sets-of-sets equivariant architecture critically impacts accuracy. This architecture allows track points to interact with other points both within their track and across their image, enabling effective error handling while maintaining symmetries across different viewpoints. We adopted this architecture from ESfM because it is particularly well-suited to our data structure.
>
> **W3. Statistical Significance:**
>
> If we understand correctly, the reviewer is referring to the results on the Strecha and BlendedMVS datasets. We believe that such low rotational errors demonstrate near-perfect performance, indicating that our method is on par with state-of-the-art methods on these datasets.

---

### Author Response · Authors · 2024-11-30
**Rebuttal by Authors**

We deeply appreciate the reviewers for their insightful feedback!

**We would like to emphasize the importance of our contribution:**

**Significance:**
The removal of outlier matches is a critical step in Structure-from-Motion (SfM). Methods like ESFM (Moran et al., 2021) and GASFM (Brynte et al., 2023), which do not effectively handle outliers, have been applied only in limited scenarios—such as the outlier-free Olsson’s dataset (Olsson & Enqvist, 2011), which was captured with a single camera, small motion, and no light changes. When these methods are applied to more challenging datasets, such as those consisting of internet photos, they result in poor accuracy. Our work introduces an effective, deep network-based approach to outlier removal in a global SfM setting, addressing this critical gap.


**Novelty:**
We have introduced the novel task of multi-view outlier removal and will make our benchmark and datasets publicly available. We believe future research can build on our work to further improve performance. Similar to the advancements seen in classical global methods, which culminated in GLOMAP (Pan et al., 2024) being on par with the state-of-the-art incremental COLMAP, we hope our contribution will serve as a foundation for future progress.


**Approach:**
Our goal is to simultaneously recover both camera poses and 3D structure. To handle outliers in this context, we integrate an inlier/outlier classification module within a sets-of-sets permutation-equivariant architecture. This architecture efficiently manages point tracks while respecting their symmetries, thus eliminating the need for data augmentation. We further explain the details of this equivariant architecture below.


**Importance of Equivariance:**
Our network classifies individual track points as inliers or outliers by considering both their track and image memberships. The sets-of-sets-equivariant architecture allows the hidden layers for each track point to interact with other points both within and across cameras. This eliminates the need for exhaustive augmentation processes. Our comparison to a PointNet-based architecture in Table 7 demonstrates the importance of leveraging sets-of-sets equivariance in improving performance.


**Results:**
Our results demonstrate that our method can efficiently handle challenging datasets, including those with hundreds of internet photos, achieving great accuracy. In contrast, existing deep learning methods like ESFM and GASFM fail to handle such datasets due to their inability to remove outliers effectively. Other robust deep learning-based methods also struggle with large-scale datasets of internet photos. We believe this marks an important step in advancing deep learning as a leading technique for large-scale SfM.





**We have made the following revisions in the text:**

1) Thanks to the insightful question from reviewer oVEX, we identified a bug in the GLOMAP code. Specifically, in some reconstructions, unregistered cameras (cameras without any associated 3D points) were not filtered out. This oversight negatively impacted GLOMAP’s performance.
We have corrected this issue and updated the GLOMAP results, which affected the 1DSfM and MegaDepth datasets (Tables 1, 2, 9, and 10). While the updated results show an improvement for GLOMAP, the overall conclusions of our comparisons remain unchanged.


2) The best-performing method is now marked in **bold and underlined**,  we hope this makes the tables clearer for readers.


3) At the request of the reviewers, we have added the mean errors to Tables 1, 2, 9, and 10 for the 1DSfM and Megadepth datasets to enhance clarity.

---

### Meta-Review · Area_Chair_ysVS · 2024-12-23

**Metareview:**

## Summary
The paper presents a new architecture for Multiview Structure from Motion (SfM) that improves the recovery of camera pose and 3D scene structure from large, uncontrolled image collections with outliers. The architecture incorporates an outlier classification module and a robust bundle adjustment step, achieving good accuracy and competitive runtimes against contemporary deep-learning methods and classical techniques.

##Strengths
* Integrates a new inlier-outlier classification branch and a robust structure from motion (SfM) mechanism.
* Extensive experiments conducted over various datasets to prove claims, demonstrating the efficacy of these components.
* The proposed inlier-outlier prediction head shows improvement compared to ESFM.
* Uses unsupervised reprojection losses for easy fine-tuning for different scenes.
* The paper is well-presented with a comprehensive literature review and extensive experiments.

## Weaknesses
* The ablation study section needs more consideration, as it should focus on why error handling designs are better than proving the base model is better.
* Concerns arise regarding the reliability of the results, as the rotational metrics are reported in degrees, not radians.
* The authors did not use Mast3R [2] as a baseline method due to its limitations in working with large number of images.
* The paper's contribution is largely built on existing network architecture, with the addition of outlier classification output channels.
* The proposed solution for handling outliers incurs additional overhead, lacks theoretical justification, comprises inliers recall, and can not reject outliers well.

## Conclusions

Based on the reviews and author's feedback, the paper shows an interinting approach and should be accepted as most of the concerns were addressed by the authors.

**Additional Comments On Reviewer Discussion:**

There were two reviewers that did not participate in the discussion period; whereas the last review were satisfied with the answers provided by the authors.

---

### Decision · Program_Chairs · 2025-01-22

Accept (Poster)